

# Autogenic versus allogenic controls on the evolution of a coupled fluvial megafan/mountainous catchment system: numerical modelling and comparison with the Lannemezan megafan system (Northern Pyrenees, France)

Margaux Mouchené[1], Peter van der Beek[1], Sébastien Carretier[2,3], Frédéric Mouthereau[2]

[1]Univ. Grenoble Alpes, CNRS, ISTerre, F-38000 Grenoble, France
[2]GET, Observatoire Midi Pyrénées, Université de Toulouse, CNRS, IRD, 14 avenue E. Belin, F-31400 Toulouse, France
[3]Department of Geology, FCFM, Universidad de Chile, Santiago, Chile

*Correspondence to*: Margaux Mouchené (margaux.mouchene@univ-grenoble-alpes.fr)

**Abstract.** Alluvial megafans are sensitive recorders of landscape evolution, controlled by autogenic processes and allogenic forcing and influenced by the coupled dynamics of the fan with its mountainous catchment. The Lannemezan megafan in the northern Pyrenean foreland was abandoned by its mountainous feeder stream during the Quaternary and subsequently incised, leaving a flight of alluvial terraces along the stream network. We explore the relative roles of autogenic processes and external forcing in the building, abandonment and incision of a foreland megafan using numerical modelling and

compare the results with the inferred evolution of the Lannemezan megafan. Autogenic processes are sufficient to explain the building of a megafan and the long-term entrenchment of its feeding river at time and space scales that match the Lannemezan setting. Climate, through temporal variations in precipitation rate, may have played a role in the episodic pattern of incision at a shorter time-scale. In contrast, base-level changes, tectonic activity in the mountain range or tilting of the foreland through flexural isostatic rebound appear unimportant.

**1 Introduction**

Alluvial fans are prominent geomorphic objects of remarkably conical shape, constructed by the accumulation of sediments at the outlet of mountain valleys. They occupy a key position in the sediment routing system and, as such, have been widely used as recorders of external forcing on landscape evolution in a variety of settings. Controls on the building and incision of these deposits, through alternating phases of aggradation and erosion, have been shown to be related to climatic changes

(Barnard et al., 2006; Arboleya et al., 2008; Assine et al., 2014), tectonic activity (DeCelles and Cavazza, 1999), base-level oscillations (Harvey, 2002) or to a combination of those factors (Abrams and Chadwick, 1994; Dade and Verdeyen, 2007; Schlunegger and Norton, 2014). Laboratory experiments reproducing alluvial fan dynamics have helped understanding the respective roles of these controls on the fan morphology, facies changes and cyclic erosion/deposition processes (Kim and Muto, 2007; Nicholas et al., 2009; Rohais et al., 2011; Guerit et al., 2014). Both analog and numerical modelling studies





have shown evidence for autogenic processes that could be of critical importance in fan evolution (Humphrey and Heller, 1995; Coulthard et al., 2002; Nicholas and Quine, 2007). Temporary sediment storage in the fan results in cyclic behaviour, with alternating phases of deposition and incision in the absence of external forcing (eg. Coulthard et al., 2002). This behaviour is expressed in the thresholds (in runoff, slope or shear stress) defined and implemented in the models: a critical

value must be reached and exceeded for transport to be effective; after some further time steps, this parameter value decreases below the threshold and deposition occurs again (Schumm, 1979; Roering et al., 1999; Whipple and Tucker, 1999; DiBiase and Whipple, 2011).

Another level of complexity, often overlooked in previous experiments of alluvial systems, comes from the strong coupling and feedbacks between the source catchment and the basin. Specific response amplitude and time of each part of the system

to a given forcing may differ and this results in a complex, oscillating erosion signal (Densmore et al., 2007; Humphrey and Heller, 1995; Babault et al., 2005; Carretier and Lucazeau, 2005). Numerical modelling by Pepin et al. (2010) suggested that autogenic processes play a key role in the evolution of such a coupled system submitted to constant external forcing. For these authors, permanent autogenic entrenchment can occur in a coupled catchment-fan system without changes in boundary conditions and external forcing when (i) the transport threshold (critical shear stress) is significant and (ii) progradation is

limited by an open boundary with fixed elevation (e.g. a large river system at the foot of the fan).

In the northern foreland of the Pyrenees (France), the Lannemezan megafan was built since the Miocene by the erosional products of a mountainous catchment, and was abandoned during the Quaternary (Mouchené et al., submitted). The respective roles of climate and tectonics in this evolution remain unresolved.

In this study, we seek to test hypotheses on the mechanisms at play in the abandonment and incision of the Lannemezan

megafan through numerical modelling of alluvial megafan construction and abandonment. Although the complexity of this natural case might not be fully reproduced by the numerical model, we will explore trends and patterns of incision (time and space scales, amplitudes) to understand the effect of different potential forcing factors (climate, tectonics, base level change, etc.) on landscape evolution. Disentangling the respective signals of autogenic processes and allogenic forcing requires understanding of (i) the wavelength and amplitude of each signal, (ii) the possible buffering effects of the response times of

the fan and of the mountainous catchment, and (iii) the amplification/reduction factors introduced by the coupling of the system.

## 2 The Lannemezan megafan

Whereas the drainage network in the Pyrenean range is regularly spaced and mostly transverse to the structural trend, rivers of the northwestern foreland spread in a radial pattern over the convex topography of large Miocene alluvial fans (Fig. 1).

The Lannemezan megafan is the most prominent geomorphic feature of the northern Pyrenean foreland, with a surface of 13,000 km$^2$ and a mean slope of 0.3°. Its characteristic semi-conical shape is outlined by the Garonne river (to the south, east and north) and by the radial river network on its surface.





Molasse-type deposits of middle-late-Miocene age, with rounded pebbles and boulders in an abundant, clayey and sandy matrix, make up most of the megafan volume (Paris, 1975; Azambre et al., 1989). These are capped by the *Lannemezan Formation,* consisting of (i) a well-sorted, stratified clay and sand sequence, containing strongly weathered gravel and pebbles, dated by a Hipparion-bearing fauna at its base as latest Miocene to Pliocene ("Pontico-Pliocene"; Paris, 1975; Azambre et al., 1989), and (ii) a Quaternary sheet of very similar composition.

The Neste River exits the mountain range at the apex of the megafan and thus most probably provided the material building the megafan. However, this stream now turns sharply to the east at this point and incises the fan head ~100m vertically, before merging with the larger Garonne River at its mountainous outlet (Fig. 1). The capture of the Neste by the Garonne and related abandonment of the fan was dated by Mouchené et al. (submitted) at ~300ka, from [10]Be and [26]Al cosmogenic nuclide dating of the fan surface.

During incision, the rivers of the northern foreland (including the Neste, Garonne and fan rivers) left a series of alluvial terraces, the episodic abandonment of which was dated by these authors and may be related to changing fluvial dynamics during shifts between Quaternary cold and warm phases (Mouchené et al., submitted).

### 3 Model description

We use a recent version of the CIDRE code, which models landscape evolution in a continental setting (Carretier et al., 2015). We recall here the main characteristics of the code and refer the reader to Carretier et al. (2015) and references therein for further details.

At the beginning of each time step, a specified volume of water is distributed homogeneously over the cells making up the model surface. The propagation of water and sediment is performed in cascade, from the highest to the lowest cell and following decreasing elevation, to ensure mass conservation. A Multiple Flow algorithm is used to propagate the water flux to downstream cells proportionally to the slope in each direction (Murray and Paola, 1997; Coulthard et al., 2002; Carretier et al., 2009). This allows for a distributary drainage pattern to develop.

#### 3.1 Mass balance

During a time step $\partial t$, the elevation $z$ of a point of a cell changes as follows:

$$\frac{\partial z}{\partial t} = -\epsilon + D + U \tag{1}$$

where $\epsilon$ is a local erosion (detachment or entrainment) rate, $D$ is a local deposition rate and $U$ an uplift or subsidence rate. The local deposition rate $D$ is defined as:

$$D = \frac{q_s}{L} \tag{2}$$

with $q_s$ the incoming sediment flux per unit width and $L$ the transport length.





The transport length $L$ determines the proportion of incoming sediment flux that is deposited in the cell: a large $L$ results in a little deposition, as a steep slope or high water discharge would favour in natural settings. The cell outflux per unit width $q_s$ is the sum of the sediment detached from a given cell plus the sediment eroded upstream and that crossed this cell without being deposited; it is thus non-local (e.g. Tucker and Bradley, 2010). This approach is generalised for both hillslope and

river processes by specifying $\epsilon$ and $L$ in both cases.

## 3.2 Hillslope processes

The approach used by Carretier et al. (2015) is different from the "non-linear" diffusion model proposed by previous authors (Roering et al., 1999; Carretier et al., 2009; 2014). Instead, in this model, the elevation variation results from the difference between a local detachment rate and a deposition rate using Equations 1 and 2:

$$\epsilon = \kappa S \tag{3}$$

$$L = \frac{dx}{1 - (S/S_c)^2} \tag{4}$$

Where $\kappa$ is an erodability coefficient, $S$ is the steepest slope and $S_c$ is a critical slope. If the slope is steeper than $S_c$, $\epsilon$ is set such that $S = S_c$. The detachment rate is proportional to the local gradient, but the deposition rate ($q_s/L$ in Equation 2) depends on the slope and critical slope: when $S << S_c$, most of the sediment entering a cell is deposited there and when $S \sim S_c$,

$L$ becomes infinity and there is no deposition on the cell.

## 3.3 Fluvial Processes

For fluvial processes, a detachment algorithm including a threshold is used for sediment and bedrock:

$$\epsilon = K(k_t q^m S^n - \tau_c)^p \tag{5}$$

$$L = \xi q \tag{6}$$

where $K$ is an erodibility coefficient, $q$ is water discharge per unit flow width on the cell, $S$ is slope and the exponents are positive. $k_t$ is the shear stress parameter so that $k_t q^m S^n = \tau$ (shear stress) and Equation 5 takes the classic form of the excess shear-stress formula (Tucker, 2004). $\tau_c$ is the critical shear stress to be reached for clast detachment. $p$ is a positive coefficient, set to 1 in our experiments (following Lavé and Avouac, 2001). The transport length $L$ depends on particle size

and density (included in the coefficient $\xi$). This law implies that the deposition rate decreases when the water discharge per unit width $q$ increases.

For fluvial processes, the flow width $w$ can be set to the cell width $dx$ or to a river width such as:





$$w = k_w Q^{0.5} \tag{7}$$

where $k_w$ is a coefficient depending on the lithology and Q is the total water discharge at a river section. Flow-width variation is critical in the modelling of alluvial fan evolution because it plays a role in avulsion processes, in the changing flow dynamics (a change in flux geometry may lead to overflowing and a shift to distributive flowing), and in incision

patterns (leaving alluvial terraces in some cases).

### 3.4 Cover effect

Erosion of sediment is different from that of bedrock (Equations 3 and 5), and, within the bedrock, different layers can be defined by their respective erodibility and detachment or slope thresholds ($\kappa$ and $S_c$ for hillslope processes and $K$ for fluvial processes). During a time step $dt$, different layers can be eroded on a given cell: the erosion of each layer consumes part of $dt$

so that less time remains to erode the underlying layer. This time reduction is taken into account by multiplying $dt$ by $(1 - \frac{volume\ layer}{w\ dx\ \epsilon dt})$ between layers. In this way, the "cover effect" of a sediment layer covering the bedrock (e.g Whipple and Tucker, 2002; Lague, 2010) can be taken into account.

### 3.5 Lateral erosion

Flowing water can erode lateral cells, which are topographically above them and placed in a lateral direction perpendicular to

each downstream direction. The lateral sediment flux $Q_{sl}$ is defined as a fraction of the flux in the considered direction (e.g. Murray and Paola, 1997; Nicholas and Quine, 2007):

$$Q_{sl} = \alpha Q_s \tag{8}$$

where $\alpha$ is a bank erodibility coefficient; it is specified for sediment and implicitly determined for bedrock layers, proportionally to their erodibility (i.e. $\alpha_{sediment}/\alpha_{bedrock} = K_{sediment}/K_{bedrock}$ with $K$ from Equations 5 and 7).

**4 Model setup**

The model simulates the evolution of a 100-by-150 km region split into a foreland zone (100x100 km) and an uplifting mountain zone (100x50km; Fig. 2). The grid cell size is 500x500 m. The dimensions are chosen to allow megafan building on an area matching that of the Lannemezan megafan and to permit competing catchments to develop during the drainage network growth phase; they correspond to a compromise between computing time and spatial resolution. Our model has

much larger dimensions than previous experiments on coupled catchment-foreland systems (Tucker, 2004; Nicholas et al., 2009; Pepin et al., 2010; Langston et al., 2015) and the foreland/mountain width ratio is much higher than in previous work





(Pepin et al., 2010). The initial surface is a horizontal grid with a Gaussian elevation noise (σ=0.5 m) so we can study the system dynamics from the start of drainage network growth.

The mountain part of the model is uplifted as a block at a constant rate of 0.3 mm y$^{-1}$ (except in experiments where this is

explicitly modified, see below), which seems reasonable for the late stages of evolution of the central Pyrenees (Jolivet et al., 2007; Nguyen et al., in review). Homogeneous precipitation is applied at a constant rate over the entire model ($P$ = 1 m y$^{-1}$); this parameter is modified in some experiments (Exp. 2a, b, c).

The sides of the mountain block (southern border and southern third of the eastern and western borders) are closed; neither water nor sediment can exit the grid through these. The other boundaries are open, corresponding to transverse rivers of

fixed elevation (0m) and able to transport both sediment and water fluxes out of the grid.

We conducted a series of trial runs to adjust the relevant parameters in order to reproduce the first-order morphological traits of the northern Pyrenean foreland. In particular, the values for transport length ($L$), for the erodibilities of bedrock and sediments (respectively $k_{br}$ and $k_{all}$) and for the critical shear stress ($\tau_c$) need to be established. These parameters are critical in the relief evolution but are generally poorly constrained. Giachetta et al. (2015) provided a compilation of values for

erodibilities of a set of lithologies. However, these were used for models where the critical shear stress is zero and should be significantly different (approximately one to two orders of magnitude larger) when $\tau_c>0$. Also, the erodibility coefficient depends on the value of $m$ (e.g., Carretier et al., 2009) and we used a different value than that of Giachetta et al. (2015). In any case, the sediment erodibility should be larger than that of the bedrock, the ratio between the two critically influencing the landscape morphology. We thus tested this ratio and the transport length $L$ in order to reproduce the first-order

characteristics of the northern Pyrenean landscape: minimum, maximum and mean elevations of the range and foreland megafan, relief and drainage network pattern. The parameters used for the model runs are presented in Table 1.

Pepin et al. (2010) suggested that the critical shear stress should be significant for permanent incision to occur. We thus fixed a positive value for $\tau_c$ (15 Pa) following Lavé and Avouac (2001) and Pepin et al. (2010).

## 5 Results

### 5.1 Megafan building

We successfully reproduced the first-order morphology of a fluvial megafan constructed on a low-elevation, stable foreland, from the erosional products of a slowly uplifting mountain-range-like block (Fig. 3).

The drainage network initiates from the area of transition between the mountain and foreland blocks (Fig. 3 A). In the foreland, it propagates towards the north and fans aggrade. Evenly spaced rivers (every ~10 km) build small fans and

progressively lengthen their watershed towards the south through headward incision. The fans quickly merge into a *bajada*, on top of which the flow is distributive (Fig. 3 B). At around 7.65 Ma, the mountain range becomes fully connected (i.e. all cells of the mountain block are connected to the base level through the river network) and the mountain outflux is dominated



by a few large rivers (~5). In the meantime, aggradation continues in the foreland with a markedly conical pattern. The rivers situated at the easternmost and westernmost ends of the mountain range bend sharply to follow an along-strike course and quickly reach the open model boundaries, probably constrained by their short distance to a base-level outlet. In the following time steps, their watershed will increase in size by retreat of the drainage divide towards the middle of the range and the

mountainous outlets of these streams will migrate towards the nearest border (Fig. 3 C). Meanwhile, the foreland deposits are mainly provided by a single central channel, the flow of which distributes sediments largely over the whole foreland, now clearly defining a megafan (the flow spanning 180° over the foreland).

Several episodes of temporary entrenchment (< 50m) occur during the building phase. They either concern the lower parts of the fan being incised by headward incision (Fig. 3 D) or the apex being incised by the main stream (Fig. 3 E). In both cases,

within a few hundred thousand years the main stream has brought sufficient material to the entrenched zone to refill it and to overflow and become distributive again (Fig. 3 D and D'). This cyclic pattern is expected on megafans (Leier et al., 2005) and shows that the code mimics the natural fluvial dynamics of these settings.

On the long term, the mean elevation stabilizes in both the foreland and the mountain. From about 9 Ma, mean elevation stabilizes in the range (Fig. 3 B) but remains slightly positive, which means that the relief is eroded at a slower rate than the

applied uplift rate (i.e. true topographic steady state is not reached). Aggradation continues in the foreland, although at slow pace (0.015 to 0.02 mm y$^{-1}$ of mean elevation change), the time scale of aggradation is thus higher than that of the relief development. This is consistent with the observation of Babault et al. (2005) from an analog model. For them, aggradation in the foreland influences the erosion of the mountain by modifying the relative uplift rate (i.e. the difference between the uplift applied to the mountain block and the aggradation rate). Erosion of the range balances the continuously varying relative

uplift rate, creating a "dynamic equilibrium" (Babault et al., 2005). A steady-state equilibrium (in which erosion rate equals uplift rate in the mountain) cannot be reached in this landscape as long as aggradation occurs in the foreland.

**5.2 Autogenic entrenchment**

If the same conditions are maintained, natural entrenchment of the main stream occurs rapidly; over a timescale that is two orders of magnitude smaller than the fan-building timescale (Fig. 4). Contrary to the building phase, during which episodes

of temporary entrenchment occurred but were followed by refilling and overflow, the incision starting at 15.3 Ma near the apex is sufficient to constrain the main stream avulsions to the eastern half of the fan for the subsequent time steps (Fig. 4 G, H, I). A small thalweg that developed on the eastern foot of the fan episodically captures the main flow and is thus progressively deepened and incised through headward incision.

Still, the main flow remains highly distributive and overflows several times this path, before being finally permanently

captured at around 15.57 Ma (Fig. 4 J). This event triggers a rapid incision phase, reaching nearly 150 m of incision close to the apex, and larger amounts further downstream. The mean elevation change in the foreland drops dramatically upon entrenchment and the fan is subsequently being eroded (mean elevation change remains negative for the rest of the experiment, Fig. 4). Erosion sharply increases in the mountain, especially in the watershed of the now connected main





stream (Fig. 4). Figure 5 suggests that this incision leads to an increase of the relief due to rapid incision in the riverbed and little to no increased erosion on the hillslopes and ridges.

The main stream then erodes laterally its right bank in the foreland, tending towards an along-strike flow direction without further vertical incision (Fig. 4 K, L). Incision occurs in this bank and oblique to it, which eventually captures a secondary

stream of the mountain range.

### 5.3 External forcing

Subsequent experiments start from the topography obtained at the end of the "building phase" at 15.3 Ma and aim at evaluating the respective roles of different external factors on the incision pattern of the northern Pyrenean foreland. We subsequently explore the influence of changing the parameters related to climate (precipitation rate and frequency), base

level change and tectonics (uplift rate and style). These models are run for 500 ky to evidence the effects of external factors at this specific timescale, which corresponds to the abandonment and incision timescale of the Lannemezan megafan. Parameters used for these experiments are summarized in Table 1.

### 5.3.1 Precipitation rate and style

Decreasing (Exp2a1) or increasing (Exp 2a2) the precipitation rate only results in decelerating or accelerating the processes

observed in the original experiment. The same evolution is observed in experiment 2a1 as using the default settings, but the evolution is slower and the model does not reach the permanent entrenchment stage after the 500 ky simulation. In the experiment with increased precipitation rate (2a2), erosion is enhanced and results in important widening of the valleys, but scattered deposition in the lower valleys create instabilities that perturb the model results.

We set the precipitation rate of experiment 2b to follow a sinusoidal distribution with 100 ky cycles, to simulate the

Quaternary climatic cycles. In this case, the trends of mean elevation change in the mountain and in the foreland are inversely correlated (Fig. 6). The mean elevation of the foreland slightly increases through the experiment but remains stable in periods of low runoff, as the sediment supply from the mountain is halted. The mountain is eroded in periods of maximum runoff, whereas the elevation increases (at the uplift rate) in periods of minimum runoff. There is a slight delay in the mountain response to the variations in precipitation: as the runoff starts to increase, the elevation in the mountain continues

to rise at the uplift rate for another time step (10 ky) before it starts to decrease (Fig. 6). Similarly there is a small lag between maximum runoff and minimum mean elevation change (Fig. 6). This delay corresponds to the response time of the mountain to cyclic precipitation rate changes and is consistent with works by Carretier and Lucazeau (2005) and Braun et al. (2015), who suggested 1 to 30 ky offset between forcing and response to rainfall variability at Milankovitch periods. In our experiment, the same delay is observed in the foreland, although the signal is less clear for periods of high runoff (Fig. 6).

Overall, during more humid periods, the mountain is eroded and the sediments are transported to the foreland, which provides material for fan aggradation, whereas in periods of low precipitation, the material is not eroded and/or transported from the mountain to the piedmont (Fig. 6).



At the end of the 500-ky simulation, no permanent entrenchment is observed on the megafan. The small amounts of incision that occur on the fan when precipitation decreases are more than compensated by renewed sediment influx from the mountain as precipitations start to increase again. The incision of the riverbed in the mountain in periods of high runoff is more than compensated for by the uplift in periods of low runoff (dominated by the applied uplift; Fig. 6)

### 5.3.2 Base-level change

A 50-m drop in base level is applied at the beginning of experiment 3a. This leads to erosion in the foreland through headward incision of a number of thalwegs, developing mostly on the western and eastern borders and persisting until the end of the experiment. Connection between the main stream and the largest incising thalweg on the eastern border happens earlier than in the default model (at around 250 ky) but the subsequent landscape evolution is very similar in both cases, although more incised thalwegs remain at the end of this experiment (Fig. 7).

### 5.3.3 Uplift rate

Experiment 4a tests a scenario where uplift stops after the megafan building phase. In this experiment, the thalweg to the east border is continuously incised and connects with the outlet of a secondary river (at 300 ky) before connecting to the main central channel at the end of the experiment (500 ky, Fig. 8). The mean erosion rate in the range decreases steadily down to 0.19 mm y$^{-1}$ (value for last 10 ky of the experiment).

Increasing the uplift rate to 1 mm y$^{-1}$ (Experiment 4c) quickly and permanently increases the elevation in the range without increasing much the aggradation in the foreland. This may be due to erosion (detachment and/or transport) not responding rapidly enough to catch up with this increase. Permanent entrenchment occurs at the end of the experiment (500 ky) through the same process as in the default experiment.

### 5.3.4 Tilting experiment

In this experiment (Experiment 5), we seek to reproduce the effect of isostatic rebound on the erosional pattern of the range and its foreland. At the moment, the CIDRE model does not include flexure. We thus chose to simulate the first-order effect of the flexural response to erosional unloading of the range through simple linear tilting of the model. This corresponds to an uplift pattern that increases linearly from 0 at the north boundary to a maximum fixed value at the south boundary.

To scale the tilting to the observed geomorphic characteristics of the northern Pyrenean foreland, we estimate the potential tilting of the Lannemezan megafan. We use a scaling law between fan area and fan slope to estimate the initial depositional slope of the Lannemezan Formation that caps the Miocene deposits. We use this formation because its base is the only mapped surface effectively preserved from erosion since deposition. We use a digitized geological map and an ASTER DEM (70 m resolution) to extrapolate a base surface using ArcGIS software and estimate its current slope at 0.5°. Figure 9A shows area and slope data for the Lannemezan megafan compared to data compilations from active and inactive alluvial fans and megafans from the Alps, the Andes and the Himalaya (Horton and DeCelles, 2002; Guzzetti et al., 1997). The discrepancy of



the Lannemezan data with the scaling law suggests an estimated ~0.4° tilt, which will be simulated by uplift increasing linearly from 0 at the north boundary to 2 mm y$^{-1}$ at the southern boundary.

It should be noted that this scaling relationship suggests a depositional angle of ~0.1°, which is not surprising for a megafan (e.g. DeCelles and Cavazza, 1999) but is not consistent with the slope observed in the default experiment (~0.4-0.5°).

We compare this result with the tilt estimated using another oft-used scaling relationship, between the catchment area and the fan slope (e.g. Champagnac et al., 2008). Figure 9B shows area and slope data for the Lannemezan megafan compared to data compilations from active and inactive alluvial fans from the Alps (Guzzetti et al., 1997; Crosta and Frattini, 2004; Champagnac et al., 2008). The discrepancy of the Lannemezan data with this scaling law only suggests an estimated 0.13° tilt, which will be simulated by uplift increasing linearly from 0 at the north boundary to 0.68 mm y$^{-1}$ at the southern

boundary. We test both these minimum (0.13°, Experiment 5a) and maximum (0.4°, Experiment 5b) tilt estimations.

With the linearly increasing uplift, the megafan continues to grow; the mean elevation change in the foreland is steady, positive and higher than in the default experiment (>0.2 mm y$^{-1}$, including uplift). In both experiments, connexion with the headward incising thalweg and entrenchment occur (at ~280ky in the lower tilt experiment, and at ~240ky in the higher tilt experiment) that only temporarily affects this trend because, as the tilt continues, the river outflows from this path (at ~320ky

and ~260ky respectively; Fig. 10). The mean elevation change in the mountain is steady and positive (~0.25 mm y$^{-1}$ and 1.2-1.3 mm y$^{-1}$ respectively), with peaks following the capture.

Deposition in the main path causes the stream to overflow from this channel and resume distributive flow over the megafan, but instabilities in the models blur the results (Fig. 10). This suggests that, overall, tilting of the model prevents or limits permanent entrenchment.

Figure 11 compares the evolution of a north-south topographic profile across experiment 5a (0.13° tilt) and the default run. The megafan topography on this section is rather stable over the course of the default experiment. However, the megafan slope increases significantly in experiment 5a, showing that the tilt affects the megafan slope without being fully compensated by erosion.

## 6 Discussion

### 6.1 Megafan building

In the model, the main steps of the megafan building phases are: (i) foreland deposition starts with small fans that quickly merge into a *bajada*; mountain watersheds merge so only a few streams are left; (ii) the rivers situated near the boundary change their direction to reach the shortest flow path to the border and the central stream becomes the dominant provider for foreland sedimentation; and (iii) the megafan grows thanks to cyclic flow dynamics (oscillating between channelized and

distributive flow) and reaches dynamic equilibrium.

The timescale of the building phase of the megafan is long (>10 My) when compared to active megafans of similar volumes deposited during the Quaternary (e.g. in the Alps, Andes and Himalaya; (Assine et al., 2014; Fontana et al., 2014; Abrahami,





2015) but compares well to older systems (e.g. Campanian-Maastrichtian Hams Fork formation in Utah; DeCelles and Cavazza, 1999) and is consistent with the Lannemezan megafan building phase encompassing the Early-Middle Miocene to Early Pliocene (i.e. ~15 My).

The long foreland (foreland length/mountain length = 2) allows for a large fan to develop but requires the model parameters to be set in a way that allows transportation over such great distance, in particular the parameter $L$ must be large enough. This required longer L, which may be interpreted as a smaller settling rate (Davy and Lague, 2009), is consistent with the downstream fining of sediment in the Lannemezan megafan. Sediment fining is not accounted for in CIDRE.

The boundary conditions, open in the foreland and closed in the mountain, play a key role in the development and evolution of the drainage network. In particular, open boundary conditions on all three sides of the foreland allow for (i) the central river to become dominant in the sediment flux deposited in the foreland (thus creating a megafan) as the more lateral rivers rapidly adapt their course to the shortest path reaching the base-level (along-strike to reach boundaries), and (ii) conical shape to develop (contrary to Pepin et al., 2010, where cyclic boundary condition on lateral boundaries resulted in a more *bajada*-like landform). Also, open boundary conditions in the mountain would result in strike-parallel drainage, which shows that megafan building requires a relatively large range. Thus, in natural settings, transverse rivers with efficient fluvial transport (to evacuate both water and sediments) appear necessary on all sides for a river/fan system to be singled out and grow into a megafan deposit. In the northern Pyrenean foreland, the Garonne/Ariège and Adour rivers could have played this role, which suggests that they might have existed prior to the Miocene onset of the megafan building.

In our model, the absence of subsidence in the foreland may have encouraged the development of a fan covering a large area, which imposes overfilled condition of the foreland basin. High subsidence rate would have allowed thick accumulation close to the range and thus limited its northward extension (e.g. Allen et al., 2013). This hypothesis could be tested with the addition of an algorithm for flexure (Simpson, 2006; Naylor and Sinclair, 2008). Nevertheless, the overfill hypothesis may be justified by the deceleration of subsidence rates in the Pyrenean retro-foreland since Eocene (Desegaulx and Brunet, 1990; Desegaulx et al., 1990). In any case, the impact of varying subsidence rate on megafan growth and abandonment remains to be evaluated.

## 6.2 Autogenic incision

### 6.2.1 Time and space scales

The autogenic entrenchment happens around 15.57 My, which, within the framework of the Lannemezan megafan evolution, is consistent with the fan-building phase encompassing the Early-Middle Miocene to Early Pliocene and incision taking place during the Quaternary (since 300 ka; Mouchené et al., submitted).

At 15.3 My (end of "building phase"), the mean elevation in the mountain range is about 1460 m, with a maximum elevation at 3160 m near the southern border, which is consistent with the northern flank of the Pyrenees. In the foreland, a maximum elevation of 950 m is reached at the fan apex in the model, which is higher than the current elevation of the Lannemezan



megafan apex (~ 660 m), but the mean elevation of the foreland is around 270 m. Megafan shape and dimensions (area) agree between our model and the Lannemezan megafan. At this point, the watershed of the main feeding river is about 1100 km2, which is quite large when compared to the current Neste watershed (~ 750 km2). The scale of the vertical entrenchment of the river is similar in the model and in the Lannemezan case (~100-150 m near the apex)

### 6.2.2 Mechanism/necessary conditions

For Pepin et al. (2010), autogenic entrenchment occurs only if (i) progradation is limited by the open boundary with fixed elevation and (ii) the transport threshold (critical shear stress) is significant. Nicholas et al. (2009) also suggested that declining aggradation in the fan results from increasing fan area during progradation (building phase) and incision is triggered by the lack of accommodation space when boundary conditions are reached. In nature, some incised fans are linked to powerful transverse rivers (Milana and Ruzycki, 1999; Dühnforth et al., 2007; 2008) but the causality is not proven (and external forcing is demonstrated in some cases; e.g. Dühnforth et al., 2008).

In our model, entrenchment naturally occurs, but it happens long after the moment where sediments reach the model boundaries (Fig. 3 and 4). This is different from Pepin et al (2010). In their experiments, autogenic entrenchment occurred precisely when the sediment reached the free border. We suspect that this difference comes from the lateral erosion included in our modelling and absent in simulations of Pepin et al. (2010). Lateral erosion limits the incision by fostering lateral migration and channel widening. Although our modelling seems more realistic, it is difficult to compare this prediction with natural settings because boundary conditions are likely to change over time (e.g. elevation, length; Harvey, 2002) so the comparison is not straightforward.

Consistently with Pepin et al. (2010)'s or Nicholas and Quine (2007)'s findings, the autogenic entrenchment occurs in our modelling which all use a significant critical shear stress (entrainment threshold). We suspect this threshold to control part of the incision magnitude and the delay between the moment where sediment reach the free border and the moment where incision occurs. This should be further evaluated by varying the critical shear-stress in other experiments.

Van Dijk *et al*. (2009) proposed that aggradation allows for a critical slope to be reached, triggering the incision. However, in the tilting experiments (Exp.5), the fan slope reaches greater values than in the default experiment at the time of entrenchment, and entrenchment does not occur, so this parameter does not seem to constitute a threshold for entrenchment. One explanation may be that tilting fosters erosion in the mountain, with a larger incoming sediment discharge entering the foreland, which prevents incision from growing.

### 6.2.3 Incision pattern

In the model, permanent entrenchment results from (limited) incision near the apex by the feeding river and headward incision of a thalweg from the foot of the fan until both ends meet to define a continuously entrenched pathway (Fig. 4). In the case of the Lannemezan megafan, we cannot provide evidence to support or disprove this mechanism but the drainage pattern resembles the model (Fig. 12 A and B). We could therefore envisage the following scenario:



- pre-existing river Ariège/lower Garonne flowing through the foreland, Neste river feeding the Lannemezan megafan through distributive pattern (Fig. 12 C);

- tributary of Ariège/Garonne river retreat headwards toward the west (toward the apex of the megafan; Fig. 12 D)

- migration of the tributary toward the west leading to sequential capture of (1) the upper Garonne and (2) the upper
Neste, abandonment of the fan, rapid incision and terrace formation (preferably on left bank due to river migration; Fig. 12 E).

The amount of incision is already very important in the time step following the connexion (~100m near the apex) and will only be further increased by another 80 m near the apex. Downstream, the stream erodes its right bank towards a strike-parallel pathway but does not incise vertically. This last characteristic resembles the right-lateral migration of the Neste and
Garonne rivers during their incision, evidenced by the extensive alluvial terrace staircase left almost systematically on the left banks. In our model, the sediments of the channel bed could enhance the effect of lateral incision and inhibit further vertical incision (through their cover effect; see also Hancock and Anderson, 2002; Brocard and Van der Beek, 2006).

However, the terraces of the northern Pyrenean foreland prove that incision of the Lannemezan megafan (100-m at the apex) was episodic, which contrasts with the pulse of incision predicted by the model. Cosmogenic nuclide surface exposure dating
suggests that these incision episodes in the northern Pyrenean foreland are linked to cold-to-warm climatic transitions (Mouchené et al., submitted).

### 6.3 Impact of climatic change

In the model with sinusoidal precipitation rates (Experiment 2b), more humid periods are characterized by erosion in the
mountains and deposition in the foreland (with episodic incision); both decrease in drier periods because stream power decreases and less material is being transported from the mountains. The wet-to-dry transition corresponds to a decrease in sediment input but also to a decrease in fluvial efficiency as the runoff nears 0, which prevents incision.

In the northern Pyrenean foreland, incision and abandonment of alluvial terraces has been linked to cold-to-warm climatic transitions (Mouchené et al., submitted) where the rapid decrease in sediment flux and gradual transitioning of the river to a
single meandering thread, with a low width/depth ratio, would encourage vertical incision (e.g. Hancock and Anderson, 2002).

Warm-to-cold transitions can also be associated with incision because of the increase in runoff variability and decline in vegetation that characterizes these periods, but in nature, they are usually more gradual than cold-to-warm transitions. During glacial (dry, cold) periods, regolith is actively produced on hillslopes by efficient frost cracking but it is mobilized
only at the onset of the following interglacial (wetter) period, when rainfall increases (e.g. Carretier et al., 1998). To reproduce this, we would need to include a climate (temperature)-dependant law for production of sediment.

In nature, incision is not always related to the return of wetter conditions; Meyer et al. (1995) suggest that incision of the terraces in their study site in northwestern Yellowstone National Park happens during warmer, more drought-prone periods





because of the infrequent floods scouring the channel bed. Langston et al. (2015) recently modelled a similar pattern of incision by applying more intense, longer duration precipitation events during interglacial periods, but without changing the average precipitation rate. Periglacial processes have also been suggested to be a key controlling factor for erosion (e.g. Dühnforth et al. 2010; Dosseto and Schaller, 2016): erosion is enhanced during cold periods in regions where they occur; it

is enhanced during warmer periods in regions exempt of periglacial processes. Mass wasting processes could be the main driver for erosion increase during wet periods (e.g. Bookhagen et al. 2005b) although their relationship to other environmental parameters, such as vegetation cover, remains disputed (e.g. Istanbulluoglu and Bras, 2005; Carretier et al., 2013; Dosseto and Schaller, 2016). Our current model does not to take into account such processes. Aggradation and incision thus seem to be controlled by the variability in rainfall intensity and event duration but also by temperature-

dependent hillslope processes, rather than by mean precipitation rate.

In nature, a number of studies relate terrace incision with climatic changes (Barnard et al., 2006; Bridgland and Westaway, 2008). This also seems to be the case in the northern Pyrenean foreland, where terrace abandonment was related to Quaternary climatic changes, although the model does not reproduce this pattern (it does not produce terraces at all). Several experiments suggest that the longer the foreland, the more it buffers the effects of short-period variations (Métivier and

Gaudemer, 1999; Babault et al., 2005; Carretier and Lucazeau, 2005), so the effect of rapid climatic changes could be dampened by the large dimensions of the foreland in our model, preventing terrace formation. The lack of temperature-dependent processes in our experiments (glacial erosion, temperature-dependent regolith production) may also prevent terrace formation. Finally, the model resolution could be insufficient to resolve alluvial terraces.

### 6.4 Uplift rate

In the experiment where uplift stops after 15.3 My (Experiment 4a), the mountains erode at a rate of 0.19 mm y$^{-1}$, comparable to the highest values obtained through estimation of basin-averaged erosion rates using cosmogenic nuclides in the northern Pyrenees (0.01 to 0.16 mm y$^{-1}$; Mouchené, 2016). Uplift is thought to have significantly decreased in the Pyrenees since the Miocene, with modern GPS-derived uplift rates being very small (0.1 ± 0.2 mm y$^{-1}$; Nguyen et al., in review). Our results suggest that the Lannemezan megafan could have been built in a period of reduced tectonic uplift. The

evolution of the piedmont is very similar to that of the default experiment (where uplift is maintained at 0.3 mm y$^{-1}$) except for the entrenchment that is refilled in experiment 4a. Thus, it appears that tectonic activity in the mountain belt does not strongly influence incision dynamics in the foreland.

### 6.5 Flexural isostatic rebound

We attempted to simulate the effect of flexural isostatic rebound on the incision pattern through tilting of the model. In the

Alps, tilting of the foreland is related to isostatic rebound in response to accelerated glacial erosion (Champagnac et al., 2008). This pattern was not demonstrated for the Pyrenees. Although the simplistic approach we used does not reproduce the flexural response to erosional unloading of the range in detail, the slope of the fan topographic profile increases with time





through this process, as suggested for alpine fans by Champagnac et al. (2008). Quantification of this increase in slope, although complicated by poor outcrop conditions, needs to be done in the northern Pyrenean piedmont to compare with the slope angles obtained in our model. In any case, in the experiment, tilting prevented permanent entrenchment (incision was only temporary) so this mechanism cannot explain the abandonment of a foreland megafan.

5    In the model, the topographic profiles merge downstream as a consequence of tilting. The alluvial terraces along the northern Pyrenean rivers also merge downstream and this pattern is also observed in the Alpine foreland. However, this pattern does not necessarily relate to tilting of the megafan: in other settings, this characteristic has been interpreted as a climatic imprint on incision (Poisson and Avouac, 2004; Wobus et al., 2010; Pepin et al., 2013). Thus, tilting does not appear to play a major role in the evolution of the Lannemezan megafan.

10   **7. Conclusions**

Numerical modelling of the evolution of a catchment/foreland system has provided (i) new insight in the building and incision of a foreland megafan and (ii) key elements to infer the driving forces in the natural evolution of the remarkable Lannemezan megafan and its mountainous catchment, in the northwestern Pyrenees.

For a megafan to develop, the foreland must be large enough to provide sufficient space for the fan to expand for a long 15   period of time and a lack of subsidence may help this process. The role of pre-existing transverse rivers flowing across the foreland seems to be critical in the building and incision of the megafan. They rapidly capture the closest streams exiting the range, which allows for a central mountainous stream to be singled out and to provide for most of the foreland deposits (stacked in a megafan). In the northern Pyrenean foreland, the throughgoing Adour and Garonne/Ariège rivers may have helped shaping of the Lannemezan megafan by clearing the sediment and water fluxes out of the megafan. The megafan 20   grows thanks to the autogenic oscillations between sheet-flow and channelized flow. These oscillations trigger small incisions that are subsequently overfilled and rapid lateral movement of the flow over the whole fan surface.

Permanent entrenchment of the Lannemezan megafan could thus be the result of autogenic processes through (i) progressive headward incision of a thalweg from the foot of the fan (not too far from the apex) and (ii) final and rapid incision of the apex once this thalweg has captured the feeding river at its mountainous outlet. No external forcing is needed to induce long-25   term entrenchment on the order of magnitude observed in the field (100-m vertical incision near the apex) but external factors cannot be ruled out. In particular, on a shorter time-scale, incision may have been influenced by Quaternary climatic variations as suggested by the abandonment of terrace staircases along the foreland rivers incising the Lannemezan megafan. Variations in precipitation rate alone do not seem to be sufficient to produce these episodic incision and alluviation phases and temperature-dependent hillslope processes may also be involved. In contrast, base-level changes, tectonic activity in the 30   mountain range or tilting of the foreland through flexural isostatic rebound appear unimportant.





**Acknowledgments**

This study was supported by French National Research Agency ANR (Project PYRAMID, ANR-11-BS56-0031) and forms part of MM's Ph.D. thesis funded by the French Ministry of Higher Education (MESR). ISTerre is part of Labex OSUG@2020 (ANR10 LABX56).

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





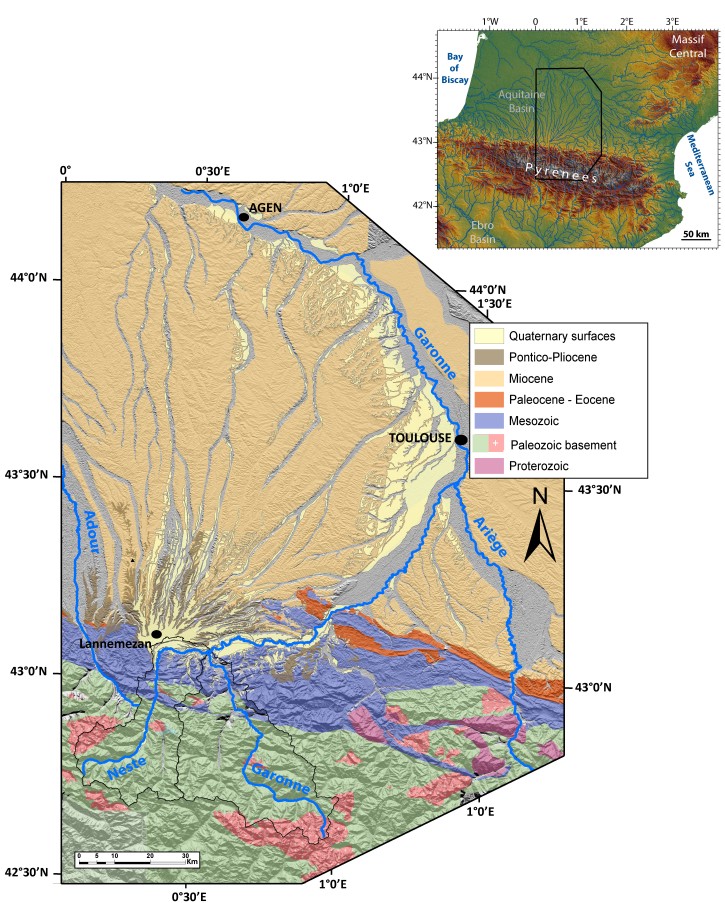

**Figure 1: The Lannemezan megafan and Neste catchment in the central northern Pyrenees (inset map shows location in southern France).**





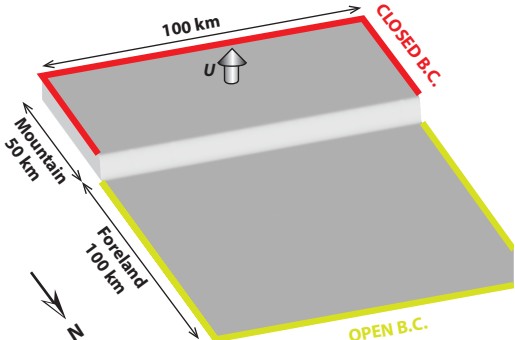

**Figure 2: The model grid consists of a mountain section, submitted to constant uplift U, and of a flat foreland section; both sections are submitted to precipitation. B.C. – Boundary Conditions; neither water nor sediment can cross a Closed B.C. while an Open B.C. corresponds to transverse rivers of fixed elevation capable of transporting both sediment and water fluxes out of the grid.**



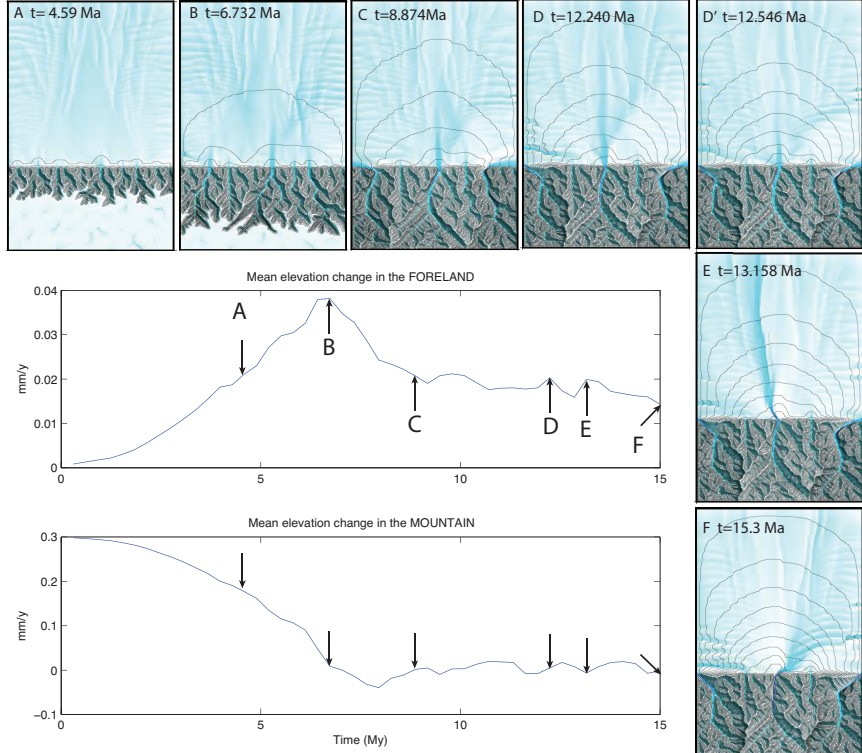

**Figure 3: Building the megafan.** Temporal evolution of the mean elevation change in the foreland and mountain and map views of the mountain and foreland landscapes (black lines are 100-m contour lines, water flux in blue shades) at the time steps marked with arrows, through the megafan building phase. **(A)** Drainage network initiates and propagates in the mountain block through headward incision while sediments are deposited along the front by regularly spaced steams. **(B)** Deposits merge in the foreland to form a *bajada* fed by a decreasing number of rivers as the mountain streams enlarge their basins. **(C)** Mean elevation in the range stabilizes and aggradation continues in the foreland, dominated by outflux of a central, main channel as the more lateral streams are drained directly toward the borders. **(D)** As aggradation continues, limited incision can occur along the borders of the fan (here on the western border) but **(D')** those thalwegs are quickly refilled. **(E)** Similarly, temporary incision can happen near the apex. **(F)** After 15.3 Ma the megafan is built.





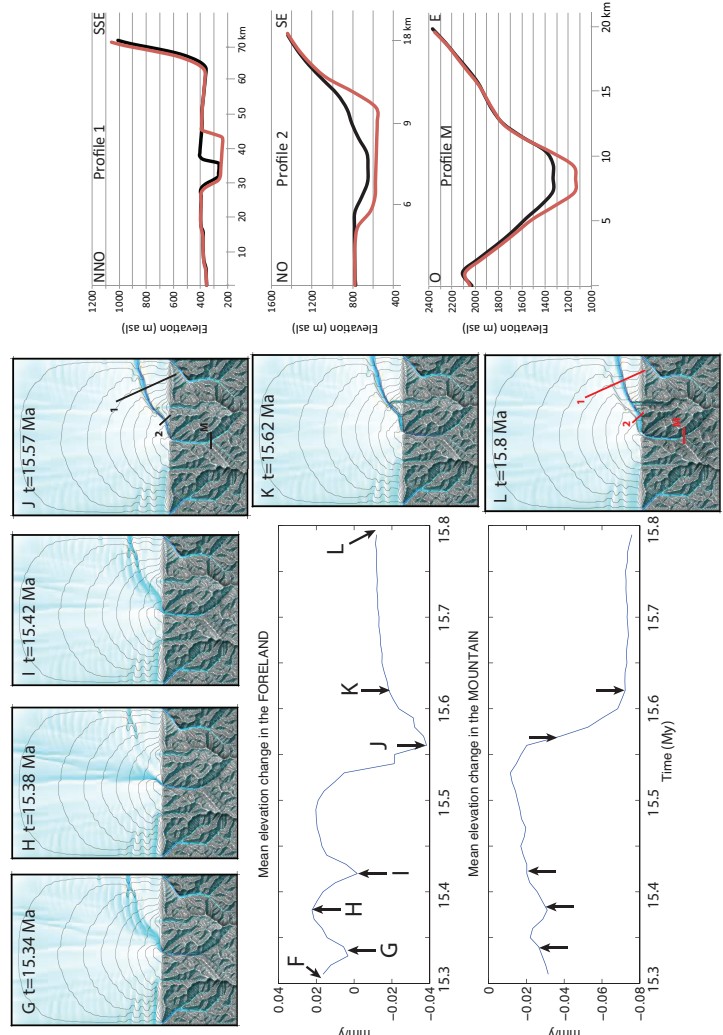

**Figure 4: Autogenic entrenchment. Temporal evolution of the mean elevation change in the foreland and map views of the mountain and foreland landscapes (black lines are 100-m contour lines, water flux in blue shades) at the time steps marked with arrows, through the autogenic incision phase. Starting from the landscape obtained at 15.3 Ma (see F of Figure 3) the fan is incised**
5 **when the main flow reaches the position of a small thalweg (e.g. G, I) but (H) continues to grow when the main flux overflows and migrates again on the fan. (J) After several of these cycles, the main flow is finally captured permanently in the thalweg. (K) As the**





main river now incises laterally towards the mountain front, a secondary stream is captured. (L) At the end of the experiment a large valley is incised along the front of the range. Topographic profiles across this valley (right panel) shows that about 120m of incision occurs in the foreland at the time of capture (black profiles, J inset); subsequently the valley is mostly enlarged by lateral erosion in the foreland, deepened and enlarged near the apex, and markedly deepened in the mountain (Note that horizontal scale 5 is different for each profile).

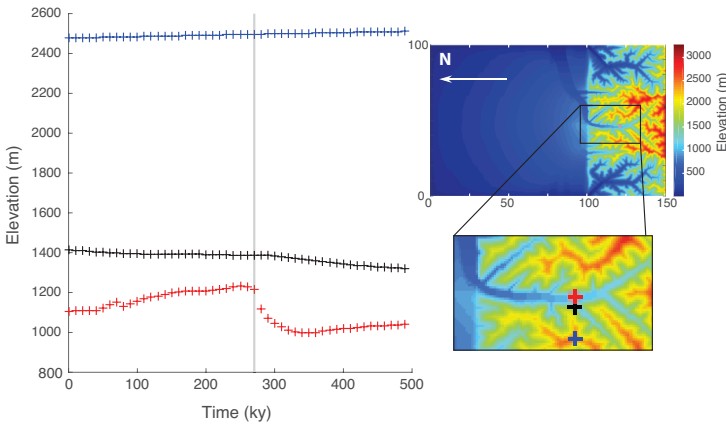

**Figure 5:** Temporal elevation change of three locations in the mountain: bed of the main feeding river (red crosses), on nearby slope (black crosses) and ridge (blue crosses, see map view in right panel for locations). Following the entrenchment (marked with 10 vertical grey line), the river rapidly incises, increasing (temporarily) the relief, as ridge elevations are not affected by the incision episode. The hillslope response is slow and lags behind that of the riverbed.



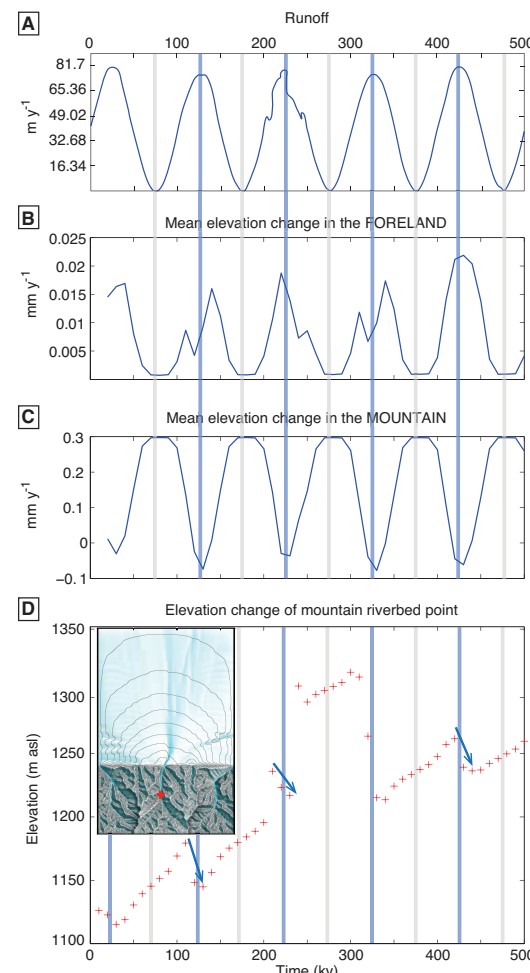

**Figure 6: Experiment 2b (sinusoidal precipitation) Temporal evolution of (A) runoff, (B) mean elevation change in the foreland and (C) in the mountain. The trends of mean elevation change in the mountain and in the foreland are inversely correlated, and show a slight delay relative to the change in runoff, corresponding to the response time. (D) Temporal elevation change of the central valley floor in the mountain (location marked by red cross on inset map) during experiment 2b. The incision (blue arrows) related to humid periods (blue vertical lines are maximum runoff) is (over-) compensated in drier periods (grey vertical lines correspond to no runoff) by the uplift, so that the elevation generally increases through the experiment. The maximum incision is delayed from the maximum runoff (response time). Temporary deposits at the valley outlet around 230 ky trigger rapid backfilling (dam) of the valley, responsible for the high elevation between 230-310 ky; the following incision episode removes this dam.**



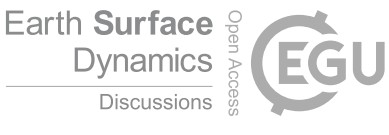

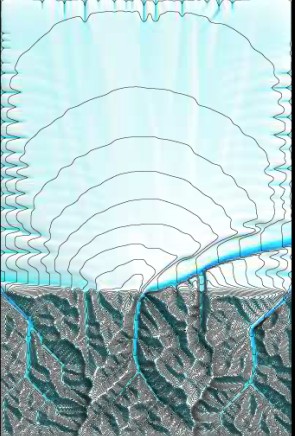

**Figure 7: Model configuration at the end (t=15.80My) of experiment 3a (initial 50-m drop in base-level). The megafan is incised by headward incision of a number of thalwegs on its western and eastern borders (and marginally on the northern border).**

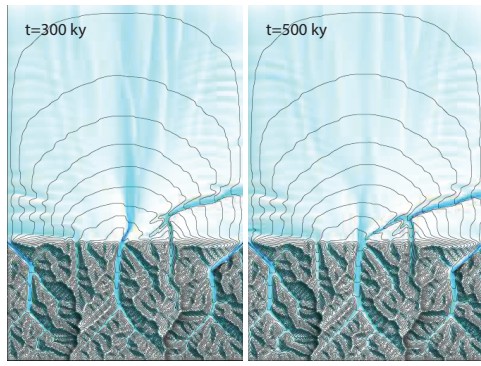

**Figure 8: Experiment 4a, the incised thalweg first connects through headward incision to the secondary river (left, t=300ky) before being connected to the outlet of the central river at the end of the experiment (right, t=500ky).**



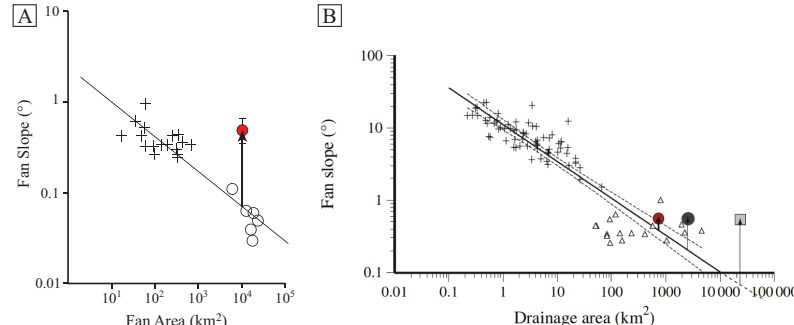

**Figure 9: (A) Scaling relationship between fan area and fan slope for alluvial systems of the Alps, the Andes and the Himalaya; data from Guzzetti et al. (1997; crosses) and Horton and DeCelles (2002; open circles). Thick line is the best power-law fit to the combined data : $S_f = 2.42\ A_f^{(-0.30)}$. Tilt (vertical arrow) for the Lannemezan megafan (red circle) is estimated as the difference between present-day slope and predicted slope from the power-law fit. (B) Scaling relationship between drainage area and fan slope for alluvial systems of the Alps, data from Guzzetti et al. (1997; crosses), Crosta and Frattini (2004; triangles) and Champagnac et al. (2008; black circle = Valensole, grey square = Chambaran). The Lannemezan megafan/Neste system (red circle) lies slightly out of the relation (fit : $S_f = 10.4\ A_b^{(-0.51\pm0.05)}$). Quaternary tilt (vertical arrow) of the Lannemezan megafan surface is estimated at 0.13° as difference between present-day slope and slope predicted by the power-law fit. Modified after Champagnac et al. (2008).**

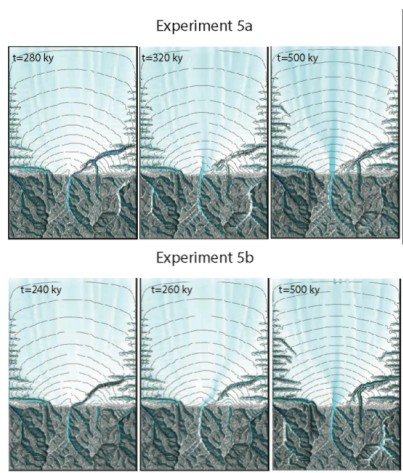

**Figure 10: Tilting experiments 5a (0.13° tilt) and 5b (0.4° tilt). Connexion with the headward incising thalweg occurs (at 280ky and 240 ky, respectively) but model instabilities in the channel, interpreted as deposition, induce overflowing (at 320ky and 260 ky, respectively). Distributive flowing over the megafan resumes and lasts until the end of the experiment.**





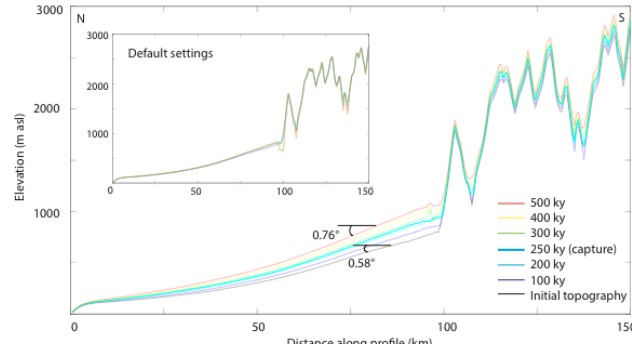

**Figure 11: Evolution of a north-south topographic profile near the middle of the model during experiment 5c (region-wide tilting). Inset shows that with default settings (experiment 1), the slope of the megafan does not increase, regional tilting (experiment 5c) is needed to create increasing northward slope through time.**



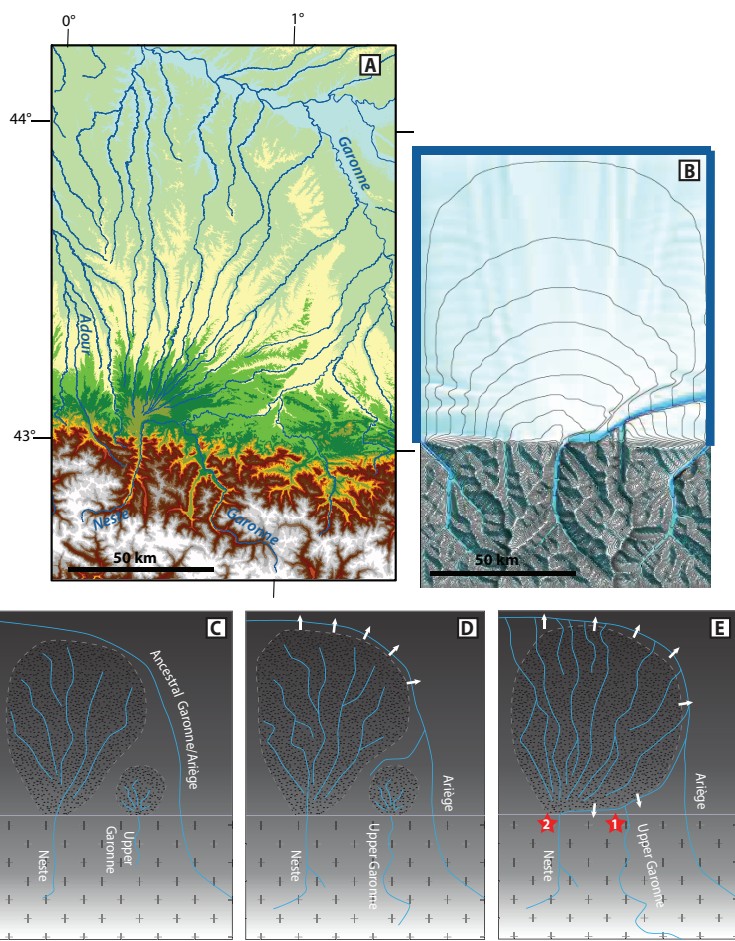

**Figure 12:** Comparison between (A) the Northern Pyrenean foreland and range DEM and (B) the final model (right; contour lines every 100 m) shown at the same scale. The proposed evolution for the Lannemezan megafan is schematized below: (C) A pre-existing river Ariège/lower Garonne flows through the foreland while the Neste transport sediments deposited in the foreland through distributive pattern to build a megafan; (D) a tributary of the Ariège/lower Garonne retreat headwards and westward, towards the apex of the Lannemezan megafan while the megafan keeps growing; (E) the migration of the tributary leads to sequential capture of (1) the upper Garonne and (2) the upper Neste and thus the abandonment of the megafan, rivers incise together in the foreland deposit leaving a series of alluvial terraces (preferably on the left bank due to the direction of river migration).



| $\tau_c$ | $K_{br}$ | $K_{all}$ | $m$ | $n$ | $p$ | $L$ | $\alpha$ |
|---|---|---|---|---|---|---|---|
| 15 Pa | $0.5 \times 10^{-3}$ | $4 \times 10^{-3}$ | 0.6 | 0.7 | 1 | 0.3 | 0.01 |

| | Settings | | | | | Results | |
|---|---|---|---|---|---|---|---|
| | Experiment number | Precipitation rate | Precipitation occurrence | Base level | Uplift rate | Permanent entrenchment | Time of entrenchment after end of building at 15.3 My |
| | | (m y$^{-1}$) | (Fraction of time step) | (m asl) | (mm y$^{-1}$) | | (ky) |
| | 1-default | 1 | 1 | 0 | 0.3 | Yes | 270 |
| Climate | 2a1 | 0.5 | 1 | 0 | 0.3 | No | - |
| | 2a2 | 2 | 1 | 0 | 0.3 | Yes | 180 |
| | 2b | 1 | sinusoidal | 0 | 0.3 | No | - |
| | 2c | 1 | 0.5 | 0 | 0.3 | Yes | 150 |
| Base Level | 3a | 1 | 1 | -50 | 0.3 | Yes | 250 |
| Uplift | 4a | 1 | 1 | 0 | 0 | Yes | 500 |
| | 4b | 1 | 1 | 0 | 0.1 | Yes | 310 |
| | 4c | 1 | 1 | 0 | 1 | Yes | 500 |
| Tilting | 5a | 1 | 1 | 0 | 0 to 0.68 | No | - |
| | 5b | 1 | 1 | 0 | 0 to 2 | No | - |

Table 1: Top: Fixed parameters for all model runs $\tau_c$ is the critical shear stress, $K_{br}$ and $K_{all}$ are the bedrock and sediment erodibility, respectively, $m$, $n$ and $p$ are coefficients for the fluvial erosion law, $L$ is the transport length and $\alpha$ is the lateral erosion coefficient. Bottom: Model settings for the experimental runs and results.