# Peer review of "Autogenic versus allogenic controls on the evolution of a coupled fluvial megafan/mountainous catchment system: numerical modelling and comparison with the Lannemezan megafan system (Northern Pyrenees, France)"

_Earth Surface Dynamics, 2016_

## Referee Comment (RC1) · Anonymous Referee #1 · 9 Sep 2016

Referee report on

"Autogenic versus allogenic controls on the evolution of a coupled fluvial megafan/mountainous catchment system: numerical modelling and comparison with the Lannemezan megafan system" by

M. Mouchené et al.

While landscape evolution models have been quite successful in realistically reproducing erosional landforms the modeling of depositional environments like alluvian fans or river deltas has shown to be hard. This difficulty can be attributed to the fact that in alluvial fans and deltas erosion and deposition closely interact with each other on the same time and length scale.

In this article Mouchené and coworkers apply the CIDRE model to study autogenic and allogenic controls and apply their simulation results to the Lannemezan megafan system in Southern France.

The paper is well written and definetly suitable for ESurf. However there are a few points, the authors should clarify before publication can be recommended.

Below my comments & suggestions for the authors:

Introduction:

As mentioned above alluvial fans are erosional/depositional environments which makes them difficult to characterize and model. The authors should emphasize this point more clearly in their introduction to distinct their work from classical landscape evolution model approaches.

Model setup:

Sentence: "We conducted a series of trial runs to adjust the relevant parameters in order to reproduce the first-order morphological traits of the northern Pyrenean foreland. In particular, the values for transport length (L), for the erodibilities of bedrock and sediments (respectively kbr and kall) and for the critical shear stress (c) need to be established."

Please clarify how the simulation results were compared to the Pyrenean foreland and describe which measures were used to determine 'similarity'. This point is also related to the following:

Results:

Sentence: "We successfully reproduced the first-order morphology of a fluvial megafan constructed on a low-elevation, stable foreland, from the erosional products of a slowly uplifting mountain-range-like block"

The authors must clearly describe what they mean by "successful reproducing the first order morphology" actually means:

1. What is the "first order morphology the authors compare their results to?s 2. Wich measure is used to determine "success"? 3. What was the initial condition which was put into the model?

See for example Topography of inland deltas, Seybold et al. GRL(2010) where the surface morphology of alluvial fans is characterized by means of simuations, lab experiments and Digital elevation model analysis and contrasted to another depositional environment specifically fluvial deltas.

Another good description of characteristics of alluvial fans can be found in: Alluvial Fans and their Natural Distinction from Rivers Based on Morphology, Hydraulic Processes, Sedimentary Processes, and Facies Assemblages, Blair et al. J. of Sedimentary Res. (1994),

p.6 L.31: What do the authors mean with "the flow is distributive" Do they mean that the system develops a diverging flow network.

Sec. 5.1 and 5.2 are rather discriptive. Do the authors have measures which characterize the formation of the fan and capture the switching of the channel network and the spatial distribution of erosion and deposition?

In sumamry the authors present a thorough description of their modeling effort but remain shallow in the quantitative interpretation of their results and a clear presentation of what they promised in the title: The constrasting of "autogenic versus allogenic controls" on the formation of depositional fans. I think the paper can make an interesting contribution if focused more clearly on the interpretation of the results using quantitative measures.

---

## Referee Comment (RC2) · H. Sinclair (Referee) · 17 Nov 2016

General comments: The Lannemezan Fan is a very interesting system to analyse, as it comprises a large radiating fan of incised channels, and forms the dominant morphological feature of the Aquitaine basin. This well presented study focuses on a numerical model using the code CIDRE in order to improve understanding of the controls on formation and subsequent incision of the fan. The model generates some interesting output which appears to approximately simulate the topography of the catchment and

a fan system that is broadly comparable to the Lannemezan system. I think there is an interesting story here in relation to controls on local base levels in continental fore-land basins and the impact on growth, stability and incision of large alluvial fans; in this case, the analysis suggests autogenic control through capture of drainage net-works has played a role in determining the form of the fan. As outlined in the specific comments below, there is a lot of scope for expanding the documentation of the Lan-nemezan fan. For example, it would be good to understand more on the timing of growth of the fan in relation to deformation of the north Pyrenean thrust wedge, and in terms of the sedimentological evolution of the system, and the evidence for tilting of its stratigraphy rather than its relatively modern terrace system. In these aspects, it is hard to judge whether the parameter combinations and boundary conditions used in the model are wholly suitable for the Lannemezan system. My recommendation is that the manuscript should be refocused primarily on the model as a series of generic experiments without aiming them at simulating one system. But that the aspects of the modelled fans be presented as examples of the autogenic controls likely to control a number of natural systems including the Lannemezan Fan. The developments of local base-levels generated by neighbouring river systems, and the implications for capture and incision of fan systems is very interesting and represents an important insight into the generic processes of alluvial fan growth in foreland basins.

P2 – L20-25 – The final paragraph of the Introduction is aimed at informing the reader of the aims and motivations for the study. The section starts by suggesting that the study will test hypotheses, but then doesn't define a hypothesis, and goes on to describe how the study will explore autogenic versus allogenic controls. This felt a bit vague, and so needs tightening up. I also think the title could reflect a more focused approach.

P3 – L3 – What is a well sorted 'sequence' in this context – are you referring to grainsize distributions in the sand, or a clear distinction of sand and mudstone beds?

P3 – If the Lannemezan Fan is the system to which the subsequent modelling is to be compared, then we need to know a bit more about the geological context of the fan,

particularly its upstream catchment, as the title suggests this is an important component of the story. It needs to be stated that the growth of the fan is all post-orogenic (I assume?), and that there is no reason to expect that the feeder catchment has changed in size over the period of growth and erosion of the fan. It would also be important to describe the glacial impact on the catchment as this will have affected the sediment and water flux. Some of the brevity in this section points to the manuscript by Mouchené et al that has been submitted. Even if that paper is published by the time this comes out, I think there is a need to summarise in more detail some of the conclusions of that study, and give more background.

Bottom P3 – On initial reading, I was unclear how L is defined – it is stated that this is the transport length, but its not clear what transport this is referring to; L is often used as transport length in diffusion models referring to catchment length (e.. Allen et al., 2013). Now having read on, I realise we are talking about a sediment transport length per unit time – this needs clarification.The parameters for the model need summarising on a figure.

P6 – L5 – The model set-up involves a fault displacement of 0.3mm/yr at the mountain front allowing for the erosion of the range while depositing in the basin. It is suggested that this is reasonable for the Lannemezan Fan – in what way? When did the uplift of the northern Pyrenean thrust wedge relative to the Aquitaine Basin cease? The model reflects syn-orogenic rather than post-orogenic boundary conditions. Given many would argue that the Pyrenees ceased active crustal thickening at around 20 Ma, doesn't the fan represent a dominantly post-orogenic setting with some normal faulting in its recent past?

P4 and 6 – The final set-up is based around modifying key parameters in order to simulate a system that broadly looks like the Lannemezan Fan. I note the other reviewer asks about how this comparison is made, and this is an interesting point. With so many parameters it is impossible for the reader to assess the role of each one and the sensitivity of the system to changes in each of them. Consequently, as is often the case, we

**ESurfD**
put a degree of faith in the model output based on a recognition that the values used for each component seem reasonable within the bounds of experiment and empirical evidence. Based on this, I raise the question of the exponents (n and m) on q and S in the fluvial erosion model which are given as 0.6 and 0.7 (Table 1); doesn't this end up with unrealistic looking river profiles? (cf. point raised by other reviewer about comparisons). Similarly, I am curious to understand how the grainsize distributions are captured in the parameter L, and how sensitive the system might be to reasonable changes in grainsize as might be influenced by the onset of glacial erosion for example.

P8 – L25-30. The tilt experiments are interesting, but the tilt values used in the experiments are very likely to represent a minimum relative to the real values. The surface they choose to evaluate whether the fan is tilted is the relatively recent surface that caps the system (I think the author dates it in her thesis at around 300 Ka). However, if the Pyrenees has been in a post-orogenic state for ca. 20 Myr, then it is reasonable to expect that the fan has been tilted much more than this in response to post-orogenic erosion. Which then raises the question of how much of the current morphology of the fan is a record of post-orogenic erosion with channels that are unrelated to the Miocene depositional system. I believe the authors have evidence to indicate that this is not the case, and so I suggest this should be discussed/demonstrated in this manuscript.

P11 –L5 – "This required longer L, which may be interpreted as a smaller settling rate (Davy and Lague, 2009), is consistent with the downstream fining of sediment in the Lannemezan megafan." Has downstream fining been demonstrated for the Lannemezan fan? This sedimentological evidence for both a radiating depositional system and documented fining rates needs to be summarised in a figure.

P11 – L20- The critical boundary condition effect here seems to be the open boundary at the sides of the basin. Firstly, there appears to be a rather unusual transverse (ie. E-W) set of channels that are regularly spaced, perpendicular to the main fan, and these seem to link to the open boundaries at the margins. What do these represent? Secondly, the lateral open boundary will always result in incision once a lateral channel

taps into the main discharge, as it represents the shortest route, and hence steepest channel, to the zero elevation boundary. It suggests that there are no along-strike fans that maintain an elevation comparable to the modelled fan. So this modelled incision due to capture by the shortest route channel is not representative of the processes of base level change along the strike of a foreland basin, unless you're next to the coast.

P12 – The point raised above seems relevant to this discussion about the controls on incision into fans and the comparison to Pepin et al's study. The open boundary which seems to be maintained at zero elevation at the margin will result in steeper channels with higher bed shear stress than those that drain out at the northern margin of the system – therefore, the former will capture the drainage of the latter, and incise into the depositional base-level built by the longer channels. This seems an inevitable consequence of these boundary conditions. The subsequent comparison to the modern channel system, and the evolutionary model proposed in figure 12 requires that the Ariege and Upper Garonne systems are at lower elevations than the Neste, causing a steepening of the lateral channels that link the two, and incision into the previous fan. Is there evidence that these were at a lower elevation? Is the outlet of these rivers lower than the Neste? I can see this as an important mechanism for drainage capture, but I don't think the default model simulations presented here are a valuable guide to the processes of capture in the Aquitaine Basin. The model boundary conditions of a zero elevation margin to the basin are not the same as the capture by a neighbouring river along the strike of a foreland basin.

P14- L20-25. Is this 'uplift' referring to differential surface uplift between the Aquitaine Basin and the Pyrenees or is it a regional uplift relative to sea-level?

P15 – L9 – "Thus, tilting does not appear to play a major role in the evolution of the Lannemezan megafan". I don't see how we can say this. The only record of tilting is from the very young terraces, whereas the fan itself is at least 20 Myr old. Is there any clear demonstration that the stratigraphy of the Lannemezan fan is not tilted?

Hugh Sinclair, Edinburgh

**ESurfD**

---

## Author Response (AR1)

Referee report on

"Autogenic versus allogenic controls on the evolution of a coupled fluvial megafan/mountainous catchment system: numerical modelling and comparison with the Lannemezan megafan system" by

M. Mouchené et al.

While landscape evolution models have been quite successful in realistically reproducing erosional landforms the modeling of depositional environments like alluvian fans or river deltas has shown to be hard. This difficulty can be attributed to the fact that in alluvial fans and deltas erosion and deposition closely interact with each other on the same time and length scale.

In this article Mouchené and coworkers apply the CIDRE model to study autogenic and allogenic controls and apply their simulation results to the Lannemezan megafan system in Southern France.

The paper is well written and definetly suitable for ESurf. However there are a few points, the authors should clarify before publication can be recommended.

Below my comments & suggestions for the authors:

Introduction:

As mentioned above alluvial fans are erosional/depositional environments which makes them difficult to characterize and model. The authors should emphasize this point more clearly in their introduction to distinct their work from classical landscape evolution model approaches.

Model setup:

Sentence: "We conducted a series of trial runs to adjust the relevant parameters in order to reproduce the first-order morphological traits of the northern Pyrenean foreland. In particular, the values for transport length (L), for the erodibilities of bedrock and sediments (respectively kbr and kall) and for the critical shear stress (c) need to be established."

Please clarify how the simulation results were compared to the Pyrenean foreland and describe which measures were used to determine 'similarity'. This point is also related to the following:

Results:

Sentence: "We successfully reproduced the first-order morphology of a fluvial megafan constructed on a low-elevation, stable foreland, from the erosional products of a slowly uplifting mountain-range-like block"

The authors must clarly describe what they mean by "successful reproducing the first order morphology" actually means:

1. What is the "first order morphology the authors compare their results to?s 2. Wich measure is used to determine "success"? 3. What was the initial condition which was put into the model?

See for example Topography of inland deltas, Seybold et al. GRL(2010) where the surface morphology of alluvial fans is characterized by means of simuations, lab experiments and Digital elevation model analysis and contrasted to another depositional environment specifically fluvial deltas.

Another good description of characteristics of alluvial fans can be found in: Alluvial Fans and their Natural Distinction from Rivers Based on Morphology, Hydraulic Processes, Sedimentary Processes, and Facies Assemblages, Blair et al. J. of Sedimentary Res. (1994),

p.6 L.31: What do the authors mean with "the flow is distributive" Do they mean that the system develops a diverging flow network.

Sec. 5.1 and 5.2 are rather discriptive. Do the authors have measures which characterize the formation of the fan and capture the switching of the channel network and the spatial distribution of erosion and deposition?

In sumamry the authors present a thorough description of their modeling effort but remain shallow in the quantitative interpretation of their results and a clear presentation of what they promised in the title: The constrasting of "autogenic versus allogenic controls" on the formation of depositional fans. I think the paper can make an interesting contribution if focused more clearly on the interpretation of the results using quantitative measures.
* * *
Earth Surf. Dynam. Discuss.,
doi:10.5194/esurf-2016-44-AC1, 2016

We thank the reviewer for their encouraging comments, careful criticism of the text and insightful remarks that have helped us clarify and improve the manuscript. We give detailed responses to specific comments below.

The reviewer asked to " clarify how the simulation results were compared to the Pyrenean foreland and describe which measures were used to determine 'similarity' ". We

compared the first order morphology of the landscape produced by the model to that of the Northern Pyrenean landscape: in both the foreland and the mountain we assessed the mean elevation, highest elevation, river spacing, relief (as valley-to-ridge elevation difference); in the foreland we also assessed the length, width and northward slope of the megafan. All these parameters agree between the modeled landscape and DEM of the northern central Pyrenees within 30 % for similarity to be accepted. This will be specified in the corrected version of the manuscript.

The reviewer asked for clarification on the use of the term " distributive " to describe the flow dynamics over the megafan. We should indeed precise that the flow on the fan is alternatively, both in space and time, (1) channelized with multiple, rapid avulsions occurring, and (2) unchannelized, overflowing the riverbed and diverging over the foreland. This pattern can be seen in Figure 3.

The reviewer pointed out that sections 5.1 and 5.2 are " rather descriptive ". These sections describe the processes observed during the building and subsequent abandonment and incision of the megafan, Figures 3 and 4 are provided to support these descriptions. We do provide some quantitative data on the megafan evolution in sections 5.1 and 5.2 and figure captions (mean elevation change, aggradation rates, vertical amplitude of temporary and permanent entrenchment, river spacing, timeframe). However, we acknowledge the lack of quantitative data regarding water and sediment fluxes and the spatial distribution of erosion and deposition. We will provide these data in the corrected version of the manuscript (fluxes and maps of erosion/deposition).
* * *
Earth Surf. Dynam. Discuss.,
doi:10.5194/esurf-2016-44-RC2, 2016

General comments: The Lannemezan Fan is a very interesting system to analyse, as it comprises a large radiating fan of incised channels, and forms the dominant morphological feature of the Aquitaine basin. This well presented study focuses on a numerical model using the code CIDRE in order to improve understanding of the controls on formation and subsequent incision of the fan. The model generates some interesting output which appears to approximately simulate the topography of the catchment and

a fan system that is broadly comparable to the Lannemezan system. I think there is an interesting story here in relation to controls on local base levels in continental foreland basins and the impact on growth, stability and incision of large alluvial fans; in this case, the analysis suggests autogenic control through capture of drainage networks has played a role in determining the form of the fan. As outlined in the specific comments below, there is a lot of scope for expanding the documentation of the Lannemezan fan. For example, it would be good to understand more on the timing of growth of the fan in relation to deformation of the north Pyrenean thrust wedge, and in terms of the sedimentological evolution of the system, and the evidence for tilting of its stratigraphy rather than its relatively modern terrace system. In these aspects, it is hard to judge whether the parameter combinations and boundary conditions used in the model are wholly suitable for the Lannemezan system. My recommendation is that the manuscript should be refocused primarily on the model as a series of generic experiments without aiming them at simulating one system. But that the aspects of the modelled fans be presented as examples of the autogenic controls likely to control a number of natural systems including the Lannemezan Fan. The developments of local base-levels generated by neighbouring river systems, and the implications for capture and incision of fan systems is very interesting and represents an important insight into the generic processes of alluvial fan growth in foreland basins.

P2 – L20-25 – The final paragraph of the Introduction is aimed at informing the reader of the aims and motivations for the study. The section starts by suggesting that the study will test hypotheses, but then doesn't define a hypothesis, and goes on to describe how the study will explore autogenic versus allogenic controls. This felt a bit vague, and so needs tightening up. I also think the title could reflect a more focused approach.

P3 – L3 – What is a well sorted 'sequence' in this context – are you referring to grainsize distributions in the sand, or a clear distinction of sand and mudstone beds?

P3 – If the Lannemezan Fan is the system to which the subsequent modelling is to be compared, then we need to know a bit more about the geological context of the fan,

particularly its upstream catchment, as the title suggests this is an important component of the story. It needs to be stated that the growth of the fan is all post-orogenic (I assume?), and that there is no reason to expect that the feeder catchment has changed in size over the period of growth and erosion of the fan. It would also be important to describe the glacial impact on the catchment as this will have affected the sediment and water flux. Some of the brevity in this section points to the manuscript by Mouchené et al that has been submitted. Even if that paper is published by the time this comes out, I think there is a need to summarise in more detail some of the conclusions of that study, and give more background.

Bottom P3 – On initial reading, I was unclear how L is defined – it is stated that this is the transport length, but its not clear what transport this is referring to; L is often used as transport length in diffusion models referring to catchment length (e.. Allen et al., 2013). Now having read on, I realise we are talking about a sediment transport length per unit time – this needs clarification.The parameters for the model need summarising on a figure.

P6 – L5 – The model set-up involves a fault displacement of 0.3mm/yr at the mountain front allowing for the erosion of the range while depositing in the basin. It is suggested that this is reasonable for the Lannemezan Fan – in what way? When did the uplift of the northern Pyrenean thrust wedge relative to the Aquitaine Basin cease? The model reflects syn-orogenic rather than post-orogenic boundary conditions. Given many would argue that the Pyrenees ceased active crustal thickening at around 20 Ma, doesn't the fan represent a dominantly post-orogenic setting with some normal faulting in its recent past?

P4 and 6 – The final set-up is based around modifying key parameters in order to simulate a system that broadly looks like the Lannemezan Fan. I note the other reviewer asks about how this comparison is made, and this is an interesting point. With so many parameters it is impossible for the reader to assess the role of each one and the sensitivity of the system to changes in each of them. Consequently, as is often the case, we

put a degree of faith in the model output based on a recognition that the values used for each component seem reasonable within the bounds of experiment and empirical evidence. Based on this, I raise the question of the exponents (n and m) on q and S in the fluvial erosion model which are given as 0.6 and 0.7 (Table 1); doesn't this end up with unrealistic looking river profiles? (cf. point raised by other reviewer about comparisons). Similarly, I am curious to understand how the grainsize distributions are captured in the parameter L, and how sensitive the system might be to reasonable changes in grainsize as might be influenced by the onset of glacial erosion for example.

P8 – L25-30. The tilt experiments are interesting, but the tilt values used in the experiments are very likely to represent a minimum relative to the real values. The surface they choose to evaluate whether the fan is tilted is the relatively recent surface that caps the system (I think the author dates it in her thesis at around 300 Ka). However, if the Pyrenees has been in a post-orogenic state for ca. 20 Myr, then it is reasonable to expect that the fan has been tilted much more than this in response to post-orogenic erosion. Which then raises the question of how much of the current morphology of the fan is a record of post-orogenic erosion with channels that are unrelated to the Miocene depositional system. I believe the authors have evidence to indicate that this is not the case, and so I suggest this should be discussed/demonstrated in this manuscript.

P11 –L5 – "This required longer L, which may be interpreted as a smaller settling rate (Davy and Lague, 2009), is consistent with the downstream fining of sediment in the Lannemezan megafan." Has downstream fining been demonstrated for the Lannemezan fan? This sedimentological evidence for both a radiating depositional system and documented fining rates needs to be summarised in a figure.

P11 – L20- The critical boundary condition effect here seems to be the open boundary at the sides of the basin. Firstly, there appears to be a rather unusual transverse (ie. E-W) set of channels that are regularly spaced, perpendicular to the main fan, and these seem to link to the open boundaries at the margins. What do these represent? Secondly, the lateral open boundary will always result in incision once a lateral channel

taps into the main discharge, as it represents the shortest route, and hence steepest channel, to the zero elevation boundary. It suggests that there are no along-strike fans that maintain an elevation comparable to the modelled fan. So this modelled incision due to capture by the shortest route channel is not representative of the processes of base level change along the strike of a foreland basin, unless you're next to the coast.

P12 – The point raised above seems relevant to this discussion about the controls on incision into fans and the comparison to Pepin et al's study. The open boundary which seems to be maintained at zero elevation at the margin will result in steeper channels with higher bed shear stress than those that drain out at the northern margin of the system – therefore, the former will capture the drainage of the latter, and incise into the depositional base-level built by the longer channels. This seems an inevitable consequence of these boundary conditions. The subsequent comparison to the modern channel system, and the evolutionary model proposed in figure 12 requires that the Ariege and Upper Garonne systems are at lower elevations than the Neste, causing a steepening of the lateral channels that link the two, and incision into the previous fan. Is there evidence that these were at a lower elevation? Is the outlet of these rivers lower than the Neste? I can see this as an important mechanism for drainage capture, but I don't think the default model simulations presented here are a valuable guide to the processes of capture in the Aquitaine Basin. The model boundary conditions of a zero elevation margin to the basin are not the same as the capture by a neighbouring river along the strike of a foreland basin.

P14- L20-25. Is this 'uplift' referring to differential surface uplift between the Aquitaine Basin and the Pyrenees or is it a regional uplift relative to sea-level?

P15 – L9 – "Thus, tilting does not appear to play a major role in the evolution of the Lannemezan megafan". I don't see how we can say this. The only record of tilting is from the very young terraces, whereas the fan itself is at least 20 Myr old. Is there any clear demonstration that the stratigraphy of the Lannemezan fan is not tilted?

Hugh Sinclair, Edinburgh
General comments: The Lannemezan Fan is a very interesting system to analyse, as it comprises a large radiating fan of incised channels, and forms the dominant morphological feature of the Aquitaine basin. This well presented study focuses on a numerical model using the code CIDRE in order to improve understanding of the controls on formation and subsequent incision of the fan. The model generates some interesting output which appears to approximately simulate the topography of the catchment and a fan system that is broadly comparable to the Lannemezan system. I think there is an interesting story here in relation to controls on local base levels in continental foreland basins and the impact on growth, stability and incision of large alluvial fans; in this case, the analysis suggests autogenic control through capture of drainage networks has played a role in determining the form of the fan. As outlined in the specific comments below, there is a lot of scope for expanding the documentation of the Lannemezan fan. For example, it would be good to understand more on the timing of growth of the fan in relation to deformation of the north Pyrenean thrust wedge, and in terms of the sedimentological evolution of the system, and the evidence for tilting of its stratigraphy rather than its relatively modern terrace system. In these aspects, it is hard to judge whether the parameter combinations and boundary conditions used in the model are wholly suitable for the Lannemezan system. My recommendation is that the manuscript should be refocused primarily on the model as a series of generic experiments without aiming them at simulating one system. But that the aspects of the modelled fans be presented as examples of the autogenic controls likely to control a number of natural systems including the Lannemezan Fan. The developments of local base-levels generated by neighbouring river systems, and the implications for capture and incision of fan systems is very interesting and represents an important insight into the generic processes of alluvial fan growth in foreland basins.

We thank Pr. Sinclair for his encouraging comments, careful criticism of the text and insightful remarks that have helped us clarify and improve the manuscript. We agree that more background information on the Lannemezan system should have been included and have added this to the new version of the manuscript. However, our study was initially set up – in terms of temporal and spatial scaling – to study the Lannemezan megafan in particular; we therefore have not followed the suggestion to refocus this as a generic experiment, because we have only explored a small part of the vast parameter space in that case.

P2 – L20-25 – The final paragraph of the Introduction is aimed at informing the reader of the aims and motivations for the study. The section starts by suggesting that the study will test hypotheses, but then doesn't define a hypothesis, and goes on to describe how

the study will explore autogenic versus allogenic controls. This felt a bit vague, and so needs tightening up. I also think the title could reflect a more focused approach.

We agree with the reviewer that the final paragraph did not state clearly enough the aim of the study; this has been corrected in the new version of the manuscript.

P3 – L3 – What is a well sorted 'sequence' in this context – are you referring to grainsize distributions in the sand, or a clear distinction of sand and mudstone beds?

We agree with the reviewer that the formulation was misleading, we are referring to a clear sorting of the grain size (upward fining, pebbles and boulders in a very fine matrix) and made this clearer in the revised version.

P3 – If the Lannemezan Fan is the system to which the subsequent modelling is to be compared, then we need to know a bit more about the geological context of the fan, particularly its upstream catchment, as the title suggests this is an important component of the story. It needs to be stated that the growth of the fan is all post-orogenic (I assume?), and that there is no reason to expect that the feeder catchment has changed in size over the period of growth and erosion of the fan. It would also be important to describe the glacial impact on the catchment as this will have affected the sediment and water flux. Some of the brevity in this section points to the manuscript by Mouchené et al that has been submitted. Even if that paper is published by the time this comes out, I think there is a need to summarise in more detail some of the conclusions of that study, and give more background.

We agree with the reviewer and included more information on the catchment in the revised version.

Bottom P3 – On initial reading, I was unclear how L is defined – it is stated that this is the transport length, but its not clear what transport this is referring to; L is often used as transport length in diffusion models referring to catchment length (e.. Allen et al., 2013). Now having read on, I realise we are talking about a sediment transport length per unit time – this needs clarification.The parameters for the model need summarising on a figure.

As stated page 4 line 1, L is a coefficient that "determines the proportion of incoming sediment flux that is deposited in the cell". Equation 2 shows that L has the dimension of a length (D=qs/L, with D the local deposition rate [$LT^{-1}$] and qs the incoming sediment flux per unit width [$L^2 T^{-1}$]). We refer readers to the paper by Carretier et al. (2015), which presents the model in much more detail.

P6 – L5 – The model set-up involves a fault displacement of 0.3mm/yr at the mountain front allowing for the erosion of the range while depositing in the basin. It is suggested that this is reasonable for the Lannemezan Fan – in what way? When did the uplift of the northern Pyrenean thrust wedge relative to the Aquitaine Basin cease?
The model reflects syn-orogenic rather than post-orogenic boundary conditions. Given many would argue that the Pyrenees ceased active crustal thickening at around 20 Ma, doesn't the fan represent a dominantly post-orogenic setting with some normal faulting in its recent past?

The reviewer is right in that uplift in the central Pyrenees is thought to have mostly ceased by the Miocene. The uplift is applied in the model to simulate the post-orogenic exhumation observed in the central part of the range (by thermochronological data, e.g. Jolivet et al., 2007) and the low uplift rates measured today (GPS data by Nguyen et al., 2016). It is also required to maintain sufficient relief (comparable to the observed relief in the Pyrenees) on long ($10^7$ y) timescales.

P4 and 6 – The final set-up is based around modifying key parameters in order to simulate a system that broadly looks like the Lannemezan Fan. I note the other reviewer asks about how this comparison is made, and this is an interesting point. With so many parameters it is impossible for the reader to assess the role of each one and the sensitivity of the system to changes in each of them. Consequently, as is often the case, we put a degree of faith in the model output based on a recognition that the values used for each component seem reasonable within the bounds of experiment and empirical evidence. Based on this, I raise the question of the exponents (n and m) on q and S in the fluvial erosion model which are given as 0.6 and 0.7 (Table 1); doesn't this end up with unrealistic looking river profiles? (cf. point raised by other reviewer about comparisons). Similarly, I am curious to understand how the grainsize distributions are captured in the parameter L, and how sensitive the system might be to reasonable changes in grainsize as might be influenced by the onset of glacial erosion for example.

The comparison between the model and the Lannemezan megafan is based on the first order morphology and timescale: we compare a number of physical (mean elevation, highest elevation, river spacing, relief in the range and foreland; in the foreland we also assessed the length, width and northward slope of the megafan) and temporal (duration of megafan building and incision phases) parameters from the model and the Lannemezan area DEM (ASTER GDEM2). Acceptable similarity is set at <30% difference between the two.

In the CIDRE model, the general detachment law, for both bedrock and sediments, is:

$$E = K(k_t q^m S^n - \tau_c)^p$$

The term $k_t q^m S^n - \tau_c$ is the shear stress, so that m=0.6 and n=0.7 (see for instance Tucker, 2004, ESPL). $\tau_c$ is the critical shear stress.
If $\tau_c$=0, the equation is simplified and the system described by a stream power law

$$E = Kq^m S^n$$

where parameters K, m and n "incorporate" the other parameters $k_t$ and p. In this case m=0.5 et n=1 (see Whipple and Tucker, 1999, JGR).
When $\tau_c$>0, $k_t \approx$ 0.05, p varies between 1 and 2 (depending on the abrasion process), and m and n are 0.6 and 0.7, respectively. We only evaluated cases where $\tau_c$>0. With such values of m and n, it can be expected that the long river profiles are very concave ($\theta \sim 0.8$). For technical reasons, we did not extract long river profiles on the modeled megafan but the rivers coming from the mountain range are indeed rather concave ($\theta > 0.5$). On the Lannemezan megafan, all rivers exhibit high concavity ($\theta_{mean} = 0.7$).

Grainsize distributions can be monitored in the CIDRE model but we did not explore this parameter in this study.

P8 – L25-30. The tilt experiments are interesting, but the tilt values used in the

experiments are very likely to represent a minimum relative to the real values. The surface they choose to evaluate whether the fan is tilted is the relatively recent surface that caps the system (I think the author dates it in her thesis at around 300 Ka). However, if the Pyrenees has been in a post-orogenic state for ca. 20 Myr, then it is reasonable to expect that the fan has been tilted much more than this in response to post-orogenic erosion. Which then raises the question of how much of the current morphology of the fan is a record of post-orogenic erosion with channels that are unrelated to the Miocene depositional system. I believe the authors have evidence to indicate that this is not the case, and so I suggest this should be discussed/demonstrated in this manuscript.

To evaluate whether the fan is tilted, we use the basal surface of Lannemezan Formation, which caps the Miocene deposits in the apex area, as stated in section 5.3.4. In Mouchené's thesis and Mouchené et al (in press), we indeed estimated the abandonment of this surface at ≥300 ka. The surface we use for estimating tilting of the fan is older,; the deposits at the base of the Lannemezan Formation contain fossils dated by Paris (1975) and Azambre et al. (1989) at the Miocene-Pliocene transition ("Pontien"). We use a scaling law to estimate the original depositional slope of the Pliocene deposits on top of this surface in order to retrieve the amount of tilting of this surface during the Quaternary (the tilting starts at the end of the building phase). The reviewer raises the question of how much of the morphology of the foreland is related to a Miocene depositional system as a megafan: our experiments suggest that the megafan hypothesis (megafan built in ~15 My, during the Miocene and subsequently rapidly dissected by river network following its abandonment) is plausible; we do not explore other scenarios to explain the foreland morphology (although this has been done by other authors; e.g. Bonnet et al, 2014, Réunion des Sciences de la Terre).

P11 –L5 – "This required longer L, which may be interpreted as a smaller settling rate (Davy and Lague, 2009), is consistent with the downstream fining of sediment in the Lannemezan megafan." Has downstream fining been demonstrated for the Lannemezan fan? This sedimentological evidence for both a radiating depositional system and documented fining rates needs to be summarised in a figure.

References to Crouzel (1957) and Azambre and Crouzel (1988), who documented downstream fining on the fan, have been added to the text.

P11 – L20- The critical boundary condition effect here seems to be the open boundary at the sides of the basin. Firstly, there appears to be a rather unusual transverse (ie. E-W) set of channels that are regularly spaced, perpendicular to the main fan, and these seem to link to the open boundaries at the margins. What do these represent?
Secondly, the lateral open boundary will always result in incision once a lateral channel taps into the main discharge, as it represents the shortest route, and hence steepest channel, to the zero elevation boundary. It suggests that there are no along-strike fans that maintain an elevation comparable to the modelled fan. So this modelled incision due to capture by the shortest route channel is not representative of the processes of base level change along the strike of a foreland basin, unless you're next to the coast.

Our hypothesis is not base-level change. We propose that the size and shape of the fan is determined by the pre-existing drainage network: the spacing of major drainage axes

(Garonne-Ariège to the east and Adour to the west) controls the maximum size of the fan; they limit its extension and efficiently evacuate sediments to the distant basin. They correspond to the open boundary conditions imposed in the model.

P12 – The point raised above seems relevant to this discussion about the controls on incision into fans and the comparison to Pepin et al's study. The open boundary which seems to be maintained at zero elevation at the margin will result in steeper channels with higher bed shear stress than those that drain out at the northern margin of the system – therefore, the former will capture the drainage of the latter, and incise into the depositional base-level built by the longer channels. This seems an inevitable consequence of these boundary conditions. The subsequent comparison to the modern channel system, and the evolutionary model proposed in figure 12 requires that the Ariege and Upper Garonne systems are at lower elevations than the Neste, causing a steepening of the lateral channels that link the two, and incision into the previous fan. Is there evidence that these were at a lower elevation? Is the outlet of these rivers lower than the Neste? I can see this as an important mechanism for drainage capture, but I don't think the default model simulations presented here are a valuable guide to the processes of capture in the Aquitaine Basin. The model boundary conditions of a zero elevation margin to the basin are not the same as the capture by a neighbouring river along the strike of a foreland basin.

In the model, the elevation of the foreland boundaries is fixed at 0 m, this is indeed an approximation. However, the long river profiles of the Garonne and Adour in the foreland are very flat (<0.5%) and the difference in elevation between the megafan and these rivers in the model is comparable to that of the Lannemezan area. We now discuss this point in the manuscript.

P14- L20-25. Is this 'uplift' referring to differential surface uplift between the Aquitaine Basin and the Pyrenees or is it a regional uplift relative to sea-level?

This value is a GPS-derived vertical velocity and corresponds to the differential uplift of mountain belt with respect to the foreland (the Aquitaine basin is not uplifted).

P15 – L9 – "Thus, tilting does not appear to play a major role in the evolution of the Lannemezan megafan". I don't see how we can say this. The only record of tilting is from the very young terraces, whereas the fan itself is at least 20 Myr old. Is there any clear demonstration that the stratigraphy of the Lannemezan fan is not tilted?

[revised manuscript text omitted]

Margaux Mouchene 9/1/y 15:37

Margaux Mouchene 9/1/y 15:37

Margaux Mouchene 9/1/y 15:37

Margaux Mouchene 9/1/y 15:37

Margaux Mouchene 9/1/y 15:38

Margaux Mouchene 9/1/y 15:38

Margaux Mouchene 9/1/y 15:38

Margaux Mouchene 6/12/y 11:02

Margaux Mouchene 9/1/y 15:39

Margaux Mouchene 6/12/y 11:03

Margaux Mouchene 9/1/y 15:39

east and north) and by the radial river network on its surface. It was built since the middle Miocene (Biteau et al., 2006), while orogenic activity within the Pyrenean range had already waned (Vergès et al., 2002; Sinclair et al., 2005). Molasse-type deposits of middle- to late-Miocene age, with rounded pebbles and boulders in an abundant, clayey and sandy matrix, make up most of the megafan volume (Paris, 1975; Azambre et al., 1989). These are capped by the *Lannemezan Formation,* consisting of (i) an upward-fining, stratified clay and sand sequence, containing strongly weathered gravel and pebbles in a very fine matrix, dated by a Hipparion-bearing fauna at its base as latest Miocene to Pliocene ("Pontico-Pliocene"; Paris, 1975; Azambre et al., 1989), and (ii) a Quaternary sheet of very similar composition.

The Neste River exits the mountain range at the apex of the megafan and thus most probably provided the material building the megafan. Comparing sediment volumes and thermochronology-derived exhumation rates, Mouchené (2016) showed that the relatively small Neste catchment could have exported a sufficiently large sediment flux to produce all of the megafan deposits. Contrary to the neighbouring Garonne valley, the Neste valley is believed to have mostly been exempt of glaciers during the Quaternary and no glacial features are known on the megafan itself (e.g. Delmas, 2015).

The Lannemezan megafan is currently disconnected from its source catchment and is being incised. The Neste turns sharply to the east at the megafan apex and incises the fan head ~100m vertically, before merging with the larger Garonne River at its mountainous outlet (Fig. 1); this drainage pattern suggests the capture of the Neste by the Garonne. During incision, the rivers of the northern foreland (including the Neste, Garonne and fan rivers) left a series of alluvial terraces. Mouchené et al. (in press) dated the abandonment of the fan and the onset of incision at ≥300ka, from $^{10}$Be and $^{26}$Al cosmogenic nuclide dating of the fan surface and terraces along the Neste valley. The episodic abandonment of these river terraces during incision may be related to changing fluvial dynamics during shifts between Quaternary cold and warm phases (Mouchené et al., in press).

**3 Model description**

We use a recent version of the CIDRE code, which models landscape evolution in a continental setting (Carretier et al., 2015). We recall here the main characteristics of the code and refer the reader to Carretier et al. (2015) and references therein for further details.

At the beginning of each time step, a specified volume of water is distributed homogeneously over the cells making up the model surface. The propagation of water and sediment is performed in cascade, from the highest to the lowest cell and following decreasing elevation, to ensure mass conservation. A Multiple-Flow algorithm is used to propagate the water flux to downstream cells proportionally to the slope in each direction (Murray and Paola, 1997; Coulthard et al., 2002; Carretier et al., 2009), allowing a distributary drainage pattern to develop.

**3.1 Mass balance**

During a time step $\partial t$, the elevation $z$ of a grid cell changes as follows:

Margaux Mouchene 5/1/y 12:37
Margaux Mouchene 7/1/y 10:15
Margaux Mouchene 9/1/y 15:42
Margaux Mouchene 7/1/y 10:14
Margaux Mouchene 9/1/y 15:43
Margaux Mouchene 9/1/y 15:43
Margaux Mouchene 9/1/y 15:43
Margaux Mouchene 9/1/y 15:44
Margaux Mouchene 9/1/y 15:45
Margaux Mouchene 9/1/y 15:45
Margaux Mouchene 9/1/y 15:45
Margaux Mouchene 9/1/y 15:46
Margaux Mouchene 9/1/y 15:48
Margaux Mouchene 9/1/y 15:47
Margaux Mouchene 9/1/y 15:47
Margaux Mouchene 9/1/y 15:47
Margaux Mouchene 9/1/y 15:48

$$\frac{\partial z}{\partial t} = -\epsilon + D + U \quad (1)$$

[revised manuscript text omitted]

For a megafan to develop, the foreland must be large enough to provide sufficient space for the fan to expand for a long period of time; a lack of subsidence may help this process. The role of pre-existing transverse rivers flowing across the foreland seems to be critical in the building and incision of the megafan. They rapidly capture the closest streams exiting the range, which allows for a central mountainous stream to be singled out and to provide for most of the foreland deposits stacked in the megafan. In the northern Pyrenean foreland, the through-going Adour and Garonne/Ariège Rivers may have helped shaping the Lannemezan megafan: the spacing of these pre-existing major drainage axes controls the size of the fan, they limit its extension and efficiently evacuate water and sediments out of the megafan. The megafan grows in response to the autogenic oscillations between sheet-flow and channelized flow. These oscillations trigger small incisions that are subsequently overfilled and rapid lateral movement of the flow over the whole fan surface.

Permanent entrenchment of the Lannemezan megafan could thus be the result of autogenic processes through (i) progressive headward incision of a stream from the foot of the fan (not too far from the apex) and (ii) final and rapid incision of the apex once this stream has captured the feeding river at its mountainous outlet. No external forcing is needed to induce long-term entrenchment on the order of magnitude observed in the field (100-m vertical incision near the apex) but external factors cannot be ruled out. In particular, on a shorter time-scale, incision may have been influenced by Quaternary climatic variations as suggested by the abandonment of terrace staircases along the foreland rivers incising the Lannemezan megafan. Variations in precipitation rate alone do not appear to be sufficient to produce these episodic incision and alluviation phases and temperature-dependent hillslope processes may also be involved. In contrast, base-level changes, tectonic activity in the mountain range or tilting of the foreland through flexural isostatic rebound appear unimportant factors in the abandonment of the megafan.

**Acknowledgments**

This study was supported by French National Research Agency ANR (Project PYRAMID, ANR-11-BS56-0031) and forms part of MM's Ph.D. thesis funded by the French Ministry of Higher Education (MESR). ISTerre is part of Labex OSUG@2020 (ANR10 LABX56). We thank Hugh Sinclair and an anonymous referee for constructive comments that helped clarify the manuscript.

Margaux Mouchene 9/1/y 16:36
Margaux Mouchene 9/1/y 16:36
Margaux Mouchene 9/1/y 16:36
Margaux Mouchene 9/1/y 16:36
Margaux Mouchene 9/1/y 16:36
Margaux Mouchene 9/1/y 16:37
Margaux Mouchene 7/1/y 17:40
Margaux Mouchene 9/1/y 16:37
Margaux Mouchene 9/1/y 16:37
Margaux Mouchene 9/1/y 16:37
Margaux Mouchene 9/1/y 16:37

[revised manuscript text omitted]

Margaux Mouchene 9/1/y 16:48

Margaux Mouchene 9/1/y 16:48

Margaux Mouchene 9/1/y 16:48

Margaux Mouchene 9/1/y 16:49

Margaux Mouchene 9/1/y 16:49

[Figure]

**Figure 11: Evolution of a north-south topographic profile near the middle of the model during Experiment 5 (region-wide tilting). Inset shows that with default settings (Experiment 1), the slope of the megafan does not increase; regional tilting (Experiment 5) is needed to create an increasing northward slope through time.**

Margaux Mouchene 9/1/y 16:49

Margaux Mouchene 9/1/y 16:49

Margaux Mouchene 9/1/y 16:49

Margaux Mouchene 9/1/y 16:49

Margaux Mouchene 9/1/y 16:49

Margaux Mouchene 9/1/y 16:49

[Figure]

**Figure 12: Comparison between (A) DEM of the Northern Pyrenees and foreland at the longitude of the Lannemezan megafan and (B) the final model (contour lines every 100 m) shown at the same scale. The proposed evolution for the Lannemezan megafan is schematized below: (C) A pre-existing river Ariège/lower Garonne River flows through the foreland while the Neste transports sediments deposited in the foreland through a distributive pattern to build a megafan; (D) a tributary of the Ariège/lower Garonne retreats headward and westward, towards the apex of the Lannemezan megafan while the megafan keeps growing; (E) the migration of the tributary leads to sequential capture of (1) the upper Garonne and (2) the upper Neste and thus the abandonment of the megafan; rivers incise together in the foreland deposit leaving a series of alluvial terraces (preferably on the left bank due to the direction of river migration indicated by white arrows).**

Margaux Mouchene 9/1/y 16:50

Margaux Mouchene 9/1/y 16:51

Margaux Mouchene 9/1/y 16:51

Margaux Mouchene 9/1/y 16:51

Margaux Mouchene 9/1/y 16:51

Margaux Mouchene 9/1/y 16:52

| $\tau_c$ | $K_{br}$ | $K_{all}$ | $m$ | $n$ | $p$ | $L$ | $\alpha$ |
|---|---|---|---|---|---|---|---|
| 15 Pa | 0.5 $10^{-3}$ | 4 $10^{-3}$ | 0.6 | 0.7 | 1 | 0.3 | 0.01 |

| | Settings | | | | | Results | |
|---|---|---|---|---|---|---|---|
| | Experiment number | Precipitation rate (m $y^{-1}$) | Precipitation occurrence (Fraction of time step) | Base level (m asl) | Uplift rate (mm $y^{-1}$) | Permanent entrenchment | Time of entrenchment after end of building at 15.3 My (ky) |
| | 1-default | 1 | 1 | 0 | 0.3 | Yes | 270 |
| Climate | 2a1 | 0.5 | 1 | 0 | 0.3 | No | - |
| Climate | 2a2 | 2 | 1 | 0 | 0.3 | Yes | 180 |
| Climate | 2b | 1 | sinusoidal | 0 | 0.3 | No | - |
| Climate | 2c | 1 | 0.5 | 0 | 0.3 | Yes | 150 |
| Base Level | 3a | 1 | 1 | -50 | 0.3 | Yes | 250 |
| Uplift | 4a | 1 | 1 | 0 | 0 | Yes | 500 |
| Uplift | 4b | 1 | 1 | 0 | 0.1 | Yes | 310 |
| Uplift | 4c | 1 | 1 | 0 | 1 | Yes | 500 |
| Tilting | 5a | 1 | 1 | 0 | 0 to 0.68 | No | - |
| Tilting | 5b | 1 | 1 | 0 | 0 to 2 | No | - |

**Table 1: Parameter values used in the experiments. Top: Fixed parameters for all model runs $\tau_c$ is the critical shear stress, $K_{br}$ and $K_{all}$ are the bedrock and sediment erodibility, respectively, $m$, $n$ and $p$ are coefficients for the fluvial erosion law, $L$ is the transport length and $\alpha$ is the lateral erosion coefficient. Bottom: Model settings for the experimental runs and results.**